# Molecular Evolution of Maize Ascorbate Peroxidase Genes and Their Functional Divergence

**DOI:** 10.3390/genes11101204

**Published:** 2020-10-15

**Authors:** Chunxiang Qu, Lin Wang, Yingwei Zhao, Chao Liu

**Affiliations:** 1School of Biology & Basic Medical Sciences, Medical College, Soochow University, Suzhou 215123, China; quchunxiang@suda.edu.cn (C.Q.); zhaoyingwei@suda.edu.cn (Y.Z.); 2School of Computer Science and Technology, Soochow University, Suzhou 215006, China; szwanglin@suda.edu.cn

**Keywords:** maize, ascorbate peroxidase, comparative genomic, molecular evolution, DNA microarray, specific expression, functional divergence

## Abstract

Ascorbate peroxidase (APX) is an important antioxidant enzyme. APXs in maize are encoded by multiple genes and exist as isoenzymes. The evolutionary history and functional divergence of the maize *APX* gene family were analyzed through comparative genomic and experimental data on the Internet in this paper. *APX* genes in higher plants were divided into classes A, B, and C. Each type of *APX* gene in angiosperms only had one ancestral gene that was duplicated along with the genome duplication or local (or tandem) duplication of the angiosperm. A total of eight genes were retained in maize and named *APXa1*, *APXa2*, *APXa3*, *APXb1, APXb2*, *APXc1.1*, *APXc1.2,* and *APXc2.* The *APX* genes of class A were located in the chloroplasts or mitochondria, and the class B and C genes were localized in the peroxisomes and cytoplasm, respectively. The expression patterns of eight *APXs* were different in vegetative and reproductive organs at different growth and development stages. APXa1 and APXb1 of maize may participate in the antioxidant metabolism of vegetative organs under normal conditions. APXa2, APXb2, APXc1.1, and APXc1.2 may be involved in the stress response, and APXb2 and APXc2 may participate in the senescence response. These results provide a basis for cultivating high-yield and resistant maize varieties.

## 1. Introduction

H_2_O_2_ is an important reactive oxygen species (ROS) produced in plants and has a relatively long lifespan in comparison to other ROS. H_2_O_2_ is a regulator of a multitude of physiological processes, such as stress responses, senescence, programmed cell death, and plant growth and development [1,2]. However, high concentrations of H_2_O_2_ cause damage to plants, so the H_2_O_2_ concentration must be strictly controlled in plant cells [3]. Antioxidant enzymes such as APX, GPX (glutathione peroxidase) and CAT (catalase) can all remove H_2_O_2_. APX has a high affinity for H_2_O_2_ and plays an important role in the fine regulation of H_2_O_2_ concentration [4].

APX catalyzes the conversion of H_2_O_2_ to H_2_O and O_2_ using ascorbate as the specific electron donor. APX is evolutionary descendant of APX–CCP hybrid A peroxidase [5], and it is found in higher plants, green algae [6,7], and red algae [8]. The first *APX* was generated as stromal *APX* (*sAPX*) in unicellular green algae, and that thylakoid membrane-attached APX (*tAPX*) occurred in multicellular charophytes during plant evolution [9]. A genome-wide analysis found that APXs in higher plants are encoded by genes of multigenic families and exist as isoenzymes [9,10,11]. Due to the limitations of the databases or screening conditions used by researchers, the number or sequence of *APX* genes identified in many species are different. For example, eight, six, and eight *APX* genes have been identified [10] in *Arabidopsis thaliana*, *Oryza sativa Japonica,* and *Zea mays*, respectively, while six, eight, and nine *APXs* have been identified [9] in Arabidopsis, rice, and maize, respectively. Eight *APX* genes have been identified [10,12] in the maize genome, but only six of those genes have the same nucleotide sequence. Based on the sequence alignment of the *APX* genes of multiple species, *APX* genes of angiosperms are clustered into five groups [10] or three groups [9]. One of the two extra groups is composed of Arabidopsis *APX4* and its orthologs, and the other group is composed of Arabidopsis *APX6* and its orthologs. Arabidopsis *APX4* is a thylakoid cavity protein and has no APX activity, so it is not a true *APX* gene [13]. Recent analysis of the *APX* protein structure confirmed that Arabidopsis APX4 and APX6 have deletion(s) or substitution(s) in amino acid(s), which is/are essential for APX activity [9]. Therefore, *APX4* and *APX6* of Arabidopsis and their orthologous genes are not true *APX* genes.

By analyzing the results of published studies [9,10,11,12], we determined the number and sequence of *APX* genes in Arabidopsis and rice. Rice has eight genes encoding APX isoforms: two cytoplasmic isoforms (OsAPx1 and OsAPx2) [14,15], two peroxidase isoforms (OsAPx3 and OsAPx4) [11,16], two mitochondrial isoforms (OsAPx5 and OsAPx6) [16,17,18], one chloroplast matrix isoform in the chloroplast (OsAPx7), and one thylakoid membrane-bound isoform in the chloroplast (OsAPx8) [11,16,17,18]. The *OsAPx4* gene plays an important role in the leaf senescence pathway mediated by ROS signaling. Salt stress increases the expression of *OsAP*x8 in roots of rice seedlings [17]. The expression levels of *OsAP*x*2* and *OsAPx7* are increased, whereas *OsAP*x*8* transcript accumulation is strongly suppressed in two-week-old seedlings subjected to salt stress [16]. Two cytosolic isoforms (*APX1*, *APX2*), two peroxidase isoforms (*APX3*, *APX5*), one mitochondrial and chloroplast matrix isoforms (*sAPX*) (dual targeting), and one thylakoid membrane-bound isoform in the chloroplast (*tAPX*) were identified in *Arabidopsis thaliana* [8,10,18,19,20,21]. During leaf senescence, the expression levels of these six genes are different [21]. *APX1* plays a key role in the response of *A. thaliana* to a moderate level of light and a combination of drought and heat stress [22,23].

Maize is an important crop, and the number and types of APX genes in maize are not consistent in different studies [9,10,12]; furthermore, there are few studies on the functional divergence of maize APX isoforms. Abscisic acid treatment increased the transcript levels of most of the *APX* genes except for *ZmAPX3* and *ZmAPX6*. The highest increases in transcript levels were observed for *ZmAPX2* and *ZmAPX1.1*, which displayed 6–10 fold increases. *ZmAPX1.1* and *ZmAPX2* also showed higher transcript levels under water stress conditions [12]. If the evolutionary history and orthologous genes of maize *APX* genes are clarified, the functions of orthologous genes in other species (e.g., rice) can be used to predict the functions of *APX* genes in maize. The burgeoning sets of angiosperm genome sequences provide the foundation for a host of investigations into the functional and evolutionary consequences of gene and genome duplication. At present, the genome duplication events of many species have been clarified [24,25,26,27], so it is possible to study the evolutionary history of angiosperm genes.

The functional divergence of different APX isoenzymes may be manifested in different subcellular locations, different expression sites (expression in tissues and organs), different transcription levels under stress conditions, and different affinities to substrates. The subcellular localization of APX isozymes in maize was predicted [10,12], but the results were different. At present, there are many newly developed prediction software or upgraded versions of previous software on the Internet [28,29,30,31], which can be used to predict the subcellular location of APX isoenzymes in maize. Analysis of the prediction results and the APX subcellular location information of the determined orthologous genes can more accurately predict the subcellular location of these APX isoenzymes in maize. The dbEST (Database expressed sequence tags) database compiles the expressed sequence tags of the maize *APX* genes in different tissues and organs. The PLEXdb database [32] contains the original data (gene chip research) of the genome-wide transcription maps of maize tissues at different growth and development stages and the original data of the expression pattern under different stresses provided by some researchers. The analysis of these experimental data can verify the tissue specificity of the expression and the response to stress of maize *APX* genes. The results of this study will clarify the evolutionary history of maize *APX* genes and prove their functional divergence and coordination.

## 2. Materials and Methods 

### 2.1. Screening of APX Protein Sequences

The whole-genome duplication (WGD) events are very complex in angiosperms. The period of genome duplication, the frequency of genome duplication, and the number of the genome increases (whole genome duplication or whole genome triplication) caused by each genome duplication event are different [24,25,26,27]. The researchers [24] have established a phylogenetic tree of 26 species, predominantly those with published genome sequences with known ancient whole genome duplications marked. We selected 12 representative species of genome duplication from these 26 species as the research objects of this article. The 12 species are as follows: maize (*Zea mays*), sorghum (*Sorghum bicolor*), rice (*Oryza sativa Japonica*), *Musa acuminata*, *A. thaliana*, soybean (*Glycine max*), tomato (*Solanum lycopersicum*), potato (*Solanum tuberosum*), grape (*Vitis vinifera*), *Selaginella moellendorffii*, *Physcomitrella patens*, and *Chlamydomonas reinhardtii*. In addition, we also selected an ancient angiosperm (*Amborella trichopoda*) whose genome has not undergone duplication events [25]. In order to better analyze the evolution of *APX* genes in angiosperms, we also selected two kinds of green algae (*Chlamydomonas reinhardtii*, *ostreococcus lucimarinus*), one kind of red alga (*Chondrus crispus*), and one kind of monocotyledonous plant (*Setaria italica*). The *APX* genes of these 17 species can be screened in *e!Ensembl Plants* (http://plants.ensembl.org/index.html). In the *Chara braunii* (taxid: 69332) database on the NCBI (National center for biotechnology information) website (https://blast.ncbi.nlm.nih.gov/Blast.cgi), the *APX* gene of a multicellular Chara (*Chara braunii*) was screened.

Six *APX* genes of *A. thaliana* were identified, and then their orthologs and paralogs from other 16 species were searched in *e!Ensembl Plants*. The protein sequences encoded by these genes were downloaded. For genes with multiple transcripts, we only select the transcript with the longest protein sequence and download these protein sequences. The blastp algorithm was used to search for *APX* genes from the *Chara braunii* (taxid: 69332) database in NCBI by the tAPX protein sequence from Arabidopsis. The expected value was 10, and the other parameters were default values.

The APX domains of all downloaded protein sequences were predicted by Pfam (http://pfam.janelia.org/search). The genes encoding short or domain-defective proteins were deleted.

### 2.2. Analysis of Phylogeny, Splice Site, and Duplication Events of APX Genes

A multiple sequence alignment of the APXs was performed by using ClustalX2.0 [33]. The protein-weighting matrices used for pairwise and multiple alignments parameters were BLOSUM30 and BLOSUM, respectively, and other parameters were defaults. The trees were constructed by maximum likelihood (ML) with MEGA6 [34]. The phylogeny was tested by the bootstrap method with 1000 replications. The substitution model was the Dayhoff model. The gap/missing data treatment was partial deletion, and the site coverage cutoff (%) was 70.

Exonic boundaries of the coding regions within *APX* genes were determined according to *e!Ensembl Plants*. The numbers of nucleotides (nt) for each exon and the intron phases were also determined.

The chromosome map of the *APX* genes was made by the chromosomal location information of maize, sorghum, rice, Arabidopsis, tomato, and grape. The chromosome location information of the *APX* genes was downloaded from *e!Ensembl Plants*. The duplication events that occurred by the genomic duplication of *APX* genes were analyzed using PGDD [24].

### 2.3. Analysis of Subcellular and Tissue Locations and Expression Pattern of APX Genes in Maize

The subcellular localization of maize APX isoforms was predicted by Know Predsite II [28], TargetP 1.1 [29], WoLF PSORT II [30], and CELLO v.2.5 [31]. The protein sequences of the 8 APX isozymes of maize were submitted to the above four websites for online prediction. The transcript sequences of maize *APX* genes were download from *e!Ensembl Plants*. The transcript sequences were aligned in dbEST. The genes with length >200 bp, maximum identity > 95%, and E-value < 10^−10^ were collected.

The experimental data of ZM20, ZM24, ZM30, ZM37, ZM38, and ZM54 were downloaded from PLEXdb [32]. See the literature for experimental materials and methods [32,35,36,37,38,39,40]. The average number of each gene was clustered in Cluster 3.0 [40]. The expression patterns of maize *APX* genes were analyzed in different tissues and organs at different growth and developmental stages, under infection with *Ustilagomaydis* and drought stress, and in the process of senescence.

## 3. Results

### 3.1. Identification of APX Genes

According to the information provided by the references [10], the *APX* genes in *A. thaliana* and their orthologous genes in 16 species were searched in *e!Ensembl Plants*. *APX4* and *APX6* genes in *A. thaliana* have no APX activity and are not a true *APX* gene [9,13]. Thus, *APX4*, *APX6,* and their orthologous genes were deleted. In the genome of maize, we searched 19 genes. The protein sequences encoded by eight of the 19 genes were detected by Pfam and possessed the APX domain, proving that these genes are genuine *APX* genes. These genes are consistent with those identified by the references [12]. Among the remaining 11 genes, one gene is the Arabidopsis *APX4* ortholog, one gene is the Arabidopsis *APX6* ortholog, and nine genes lack or do not have a complete APX domain. In this way, genes found in other species were also identified.

Finally, 96 APX genes were identified in 17 species, and their information is listed in Appendix A. *A. thaliana* has six APX genes. Maize and rice both have eight APX genes. Furthermore, the genes of *C. braunii* that were similar to the APX protein sequence of *A. thaliana* were also searched in NCBI. Five genes were identified in C. braunii, and their protein sequences were downloaded. A total of 101 protein sequences are stored in Appendix A.

### 3.2. Phylogenetic Analysis of APX Genes

The 101 genes from 18 species were clustered into three groups (Figure 1). The genes of unicellular photosynthetic algae (*Cyanidioschyzon merolae*, *Chlamydomonas reinhardtii*, *Ostreococcus lucimarinus,* and *Chondrus crispus*) were clustered in group A. However, the genes of *Chara braunii* (multicellular algae) were clustered in groups A and B. The genes of moss (*Physcomitrella patens*), gymnosperms (*Selaginella moellendorffii*), and angiosperms (monocotyledons and dicotyledons) were clustered in groups A, B, and C. Gymnosperms had multiple genes in each group. Genome duplication events in gymnosperms had not occurred, and multiple genes in each group were formed by local (or tandem) duplication. In each group, *Amborella trichopoda* had only one gene, while other angiosperms had multiple genes. *Amborella trichopoda* is an ancient angiosperm that evolved separately from the common ancestor of other existing angiosperms. *Amborella trichopoda* did not undergo genomic duplication [25], while other angiosperms have undergone genomic duplication [24,25,26]. These results indicate that *APX* genes in higher plants (moss, gymnosperms and angiosperms) could be classed into three major classes designated as A, B, and C. The unicellular photosynthetic algae had only class A, and multicellular algae chara had classes A and B. In each class, the ancestor of angiosperms had only one ancestral gene, and multiple genes in each species may be produced by this ancestral gene along with genomic duplication or local (or tandem) duplication.

The *APX* genes of monocotyledons (except *Musa acuminata* in class C) in each class clustered together, as did those of dicotyledons (Figure 1), which is consistent with the earlier divergence of monocotyledonous and dicotyledonous plants approximately 160 million years ago [41,42]. In class A, the five genes in maize, rice, sorghum, and millet were clustered with the three genes of *Musa acuminata* (*APXa3* branch). In addition, the eight genes of the other four monocots (except *Musa acuminata*) clustered into two branches, namely *APXa1* and *APXa2* branches, which formed sister clades. Both the dicotyledonous grape and potato had one gene, and each of other dicotyledons had two genes. These genes did not form a distinct sister clade. In class B, each monocot plants had two genes, which clustered into two branches (*APXb1* and *APXb2*) and formed a sister clade. The dicotyledon Arabidopsis and grape both had two genes, while tomato and potato both had three genes. The dicotyledon genes did not form an obvious sister clade. In class C, the three genes of *Musa acuminata* clustered together, and the genes of other monocotyledon formed two branches (*APXc1* and *APXc2*), which constituted one sister branch. Interestingly, there were three genes in maize, namely *APXc1.1*, *APXc1.2,* and *APXc2*. The first two genes were located on one branch, and the last one was on another branch. The dicotyledons genes were distributed on two branches, but no sister clade was formed.

Obviously, the following questions cannot be answered exactly based on the phylogenetic tree alone: (1) How many ancestral genes do angiosperms, especially maize have? (2) Did ancestral genes produce these genes through genome duplication or local (or tandem) duplication? (3) How did these ancestral genes originate?

### 3.3. Splice Sites of APX Genes

As seen in Figure 2, the majority of algae genes in class A had one exon. *P. paten* is a low-grade higher plant. There were multiple exons in its genes in classes A, B, and C (Figure 2, Appendix A). The genes of other higher plants had multiple exons. These results indicate that the ancient *APX* gene had no intron that appeared in the later stage of the evolutionary process.

Conserved exon/intron structures including exons with the same numbers of nucleotides as well as the conserved intron phases provide evidence for gene similarities and alternative characters independent of nucleotide or amino acid sequence [43]. In class A, most APX genes in higher plants shared linked exons of lengths 121 nt–101 nt–88 nt–68 nt–87 nt–75 nt–78 nt–105 nt and conserved phases (Figure 2). Similar phenomena were also been found in the other two classes (Appendix A): the class B genes shared 125 nt–50 nt–66 nt–49 nt–83 nt–80 nt–103 nt and conserved phases, and the C class genes shared 175 nt–66 nt–49 nt–86 (83) nt–80 nt–103 nt–59 nt and conserved phases. The comparative analysis of conserved exon/intron structures (Figure 3) revealed that the gene exon structures from classes A and B/C were completely different. The genes in classes B and C had a partially conserved exon/intron structure, which was 66 nt–49 nt–86 nt (83 nt)–80 nt–103 nt. These results prove that APX genes of higher plants can be divided into three types according to the differences in ancestral genes. Multiple genes of each species in each class evolved from their ancestral gene. The ancestral genes of classes A and B were first generated, and then the ancestral gene of class C was evolved by duplicating the ancestral gene of class B. The analysis of conserved exon/intron structures cannot determine whether each ancestral gene formed multiple genes of each class through genome duplication or local (or tandem) duplication.

### 3.4. Analysis of Chromosome Distribution and Duplication Events of APX Genes of Angiosperms

Conservative synteny analysis can provide evidence that genes are generated by genome duplication. However, WGD events of the angiosperms are very complicated. After genome duplication, the chromosomes also can be translocated, broken, and redirected [24,43,44,45]. Therefore, it is difficult to analyze the conserved synlinearity of angiosperm genes. The researchers [24] established a genome duplication database (PGDD), in which it can be verified whether genes of certain species were formed by WGD. The genes produced by genome duplication are often distributed on different chromosomes, while the genes produced by ancestral gene local (or tandem) duplication are often distributed on one chromosome in a tightly linked state [43]. We selected six species, maize, sorghum, rice, *A. thaliana*, tomato and grape, to analyze *APX* chromosomal distribution and duplication events [24]. *APXb1* and *APXb2* of maize belong to class B genes (Figure 1). They were produced by the duplication of a common ancestor of class B and are distributed on different chromosomes (Figure 4). Both rice and sorghum have two genes of class B, which are also distributed on different chromosomes. Both sorghum and maize have three genes of class A, which are distributed on different chromosomes. Rice has four genes of class A, two genes (*OsAPX5* and *OsAPX6*) of which are distributed on one chromosome and are closely linked (3.95 Mb, 3.96 Mb). Both sorghum and rice have two genes of class C, which are distributed on different chromosomes. Maize has three genes of class C, which are distributed on different chromosomes. The analysis of *APX* gene duplication events using PGDD showed that the genes in each class were generated through WGD events (except for *OsAPX5* and *OsAPX6* in rice). *OsAPX5* and *OsAPX6* in rice are tightly distributed on one chromosome, and they should be produced by local (or tandem) duplication.

Among the three dicotyledonous plants, the distribution and production of *APX* genes are similar to those of monocotyledonous plants, but they are not completely consistent (Figure 4). Both *APX* and *APX2* of tomato are distributed on one chromosome and are closely linked (0.17 Mb, 0.18 Mb). Tomato *APX3* and *3630* (41.54 Mb, 41.54 Mb) genes are also closely distributed on a chromosome. This implies that these two pairs of genes were produced by local (or tandem) duplication. However, both *APX3* and *APX5* of Arabidopsis are distributed on chromosome 4, but the distance between these two genes (16.66 Mb, 17.02 Mb) is larger than the distance between each pair in the two pairs of genes above. The total length of Arabidopsis chromosome 4 is 17.5 Mb [46], so it is clear that these two genes are located at the end of chromosome 4. Since the chromosome will undergo translocation after the genome is duplicated, the two genes in Arabidopsis are more likely to be produced through genome duplication.

### 3.5. Subcellular Localization of APX Isoenzymes in Maize

The researchers [10] used CELLO II and WoLF PSORT to predict the subcellular localization of six APXs in maize. We used a higher version of the software, namely CELLO v.2.5 and WoLF PSORT II, to make predictions. The prediction results between different versions were basically the same, except that the higher version predicted that APXc1.1 was more likely to be located in the mitochondria compared with the cytoplasm, while the lower version predicted the subcellular localization of APXc1.1 to be the opposite. The APXc1.1, APXc1.2, and APXc2 of maize were located in the cytoplasm [10,12], which is consistent with the prediction results of three kinds of software (Table 1). The APXa1 of maize was located in the chloroplasts [10,12], which is consistent with the prediction results of four kinds of software in this study (Table 1). The OsAPx8 isoform is located in the thylakoid membrane of the chloroplast [11,16]. We believed that the APXa1 of maize is located in the chloroplast and is likely to bind to the thylakoid membrane of the chloroplasts.

For APX isoforms encoded by the genes of class A, sAPX of Arabidopsis was a dual-targeting protein that can locate chloroplasts and mitochondria, while tAPX was located on the thylakoid membrane in chloroplasts [8,10,18,19,20,21]. Rice OsAPx5 and OsAPx6 are located in the mitochondria [16,17,18], OsAPx7 was located in the matrix of the chloroplast, and OsAPx8 was located in the thylakoid membrane of the chloroplast [11,16]. APXa2 and APXa3 of maize were located in mitochondria [13], and APXa2 of maize was located in chloroplasts or mitochondria [10]. In Table 1, the three softwares (CELLO v.2.5, WoLF PSORT II, KnowPredsite II) all predicted that maize APXa2 may be located in mitochondria and (or) chloroplasts, and TargetP 1.1 predicted that APXa2 may only be located in chloroplasts. Similarly, the three softwares (CELLO v.2.5, WoLF PSORT II, KnowPredsite II) also predicted that maize APXa3 may be located in mitochondria and (or) chloroplasts, but TargetP 1.1 predicted that APXa2 may only be located in mitochondria. APXa1 and APXa2 of maize and OsAPx8 and OsAPx7 of rice were orthologous genes, respectively, while APXa3 of maize and OsAPx5 and OsAPx6 of rice were orthologous genes (Figure 1). Based on the results of previous studies and our predictions, we believed that the APXa1 of maize is located in the thylakoid membrane of the chloroplast, the APXa3 of maize is located in the mitochondria, and the APXa2 of maize may be located in the mitochondria and chloroplast matrix.

The prediction results of the subcellular localization of APXb1 and APXb2 were very different using the four methods (Table 1). Previous predictions show that *APXb1* and *APXb2* in maize were located in the peroxisome and cytoplasm, respectively [12]. The *APX3* and *APX5* gene in *A. thaliana* has been localized to the peroxisome membrane [11,47,48,49]. *OsAPX3* and *OsAPX4* from rice have also been found in peroxisomal membranes [2,16]. The peroxisomal membrane-bound *APX* is a peroxisomal APX isoform, which has a catalytic domain facing toward the cytosol. The enzyme detoxifies H_2_O_2_, leaching out from peroxisomes into the cytoplasm [50]. Arabidopsis *APX3* and *APX5* and rice *OsAPX3* and *OsAPX4* genes belong to the genes of class B. Rice *OsAPX3* and *OsAPX4* are orthologous to maize *APXb1* and *APXb2* (Figure 1), respectively. The prediction results using KnowPredsite II show that both APXb1 and APXb2 of maize were located in the plasma membrane or peroxisome (Table 1). Based on previous studies and our predictions, we believed that *APXb1* and *APXb2* in maize are more likely to be distributed in peroxisomes. Therefore, the class A gene of maize was located in the chloroplast or the mitochondria, and the class B and C genes were located in the peroxisome and the cytoplasm, respectively.

### 3.6. Expression Pattern Analysis of Maize APX Genes in Different Tissues

The analysis of dbEST showed that eight *APX* genes had different expression patterns, although they were detected in the mixed samples under normal growth conditions (Appendix A). *APXa1* and *APXb1* were found only in a few tissues, while *APXb2* and *APXc1.1* and *APXc2* were detected in vegetative organs, embryos, and endosperm tissues. In addition, *APXc1.1*, *APXc1.2,* and *APXc2* were specifically expressed in the ear. *APXb1*, *APXb2*, *APXc1.2,* and *APXc2* were detected in the embryo sac.

In the ZM37 experiment from PLEXdb, the expression patterns of maize genes were examined in 60 tissues at different growth and developmental stages of maize by estimating gene transcript levels determined by the Nimblegen maize microarray [37]. The expression information of seven *APX* genes were obtained, but *APXc1.2* was not found in this microarray. Both *APXa1* and *APXa2* had only one transcript, and the remaining five genes had two or three transcripts in ZM37. Apparently, *APXa1* and *APXb1* were only specifically expressed in a few tissues (Figure 5). The expression of *APXa1* was specific to leaf (except immature leaf) and husk, and the expression of *APXb1* was specific to root, stem and shoot apical meristem, immature leaf, internode, and growing early seed, which were basically consistent with the results of EST (Expressed sequence tags) (Appendix A). However, *APXa2*, *APXa3*, *APXb2*, *APXc1.1,* and *APXc2* were expressed in a variety of tissues (Appendix A). *APXa2*, *APXc1.1-2,* and *APXc2-1* were highly expressed in 60 tissues (Figure 5). These genes were also detected in the corresponding tissues of EST. *APXa3* was significantly down-regulated in leaves and seedlings, which was consistent with the result of EST. However, it was highly expressed in many other tissues. The tissues where *APXa3* and *APXa1* were expressed were complementary. Similarly, the tissues where *APXb1* and *APXb2* were expressed were complementary (Figure 5).

In addition, different expression patterns may also appear in different transcripts of one gene. For example, the expression levels of *APXc1.1-1* and *APXc1.1-3* were lower than the transcript *APXc1.1-2* in the tissues of the early stage of seed development, husk, and leaf (Figure 5).

### 3.7. Expression Pattern of Maize Leaf APX Genes in Response to Infection with Ustilago Maydis and Drought Stress

The expression level of the *APXa3* gene was very low (Figure 6A–D), which was consistent with the result that the gene was not expressed in leaves and seedlings (Figure 5). However, the expression level of the *APXa1* gene was also low in the four treatments, which was different from the above results (Figure 5).

Compared with the control, the expression level of *APXa2* increased with the increase in the days of infection with *Ustilago maydis* (Figure 6A). The expression levels of *APXc1.1* and *APXc1.2* also increased in the late stage of infection (after 4 days), but the expression levels of *APXa1* and *APXb2* were the opposite. Similarly, *APXa2*, *APXc1.1,* and *APXc1.2* showed an increased expression level after infection, but the expression levels of *APXa1* and *APXb2* decreased (Figure 6B,C).

The expression patterns of maize seedling *APX*s in drought-tolerant and drought-sensitive cultivars were significantly different after drought treatment (Figure 6D). After drought treatment, the expression level of *APXa2* and *APXb2* increased, especially in severe drought treatment, and the expression level of *APXa1* and *APXc2* decreased in Han21 compared with the control. The *APXb2* expression level increased, and the *APXc1.1* expression level decreased in Ye478 compared with the control.

### 3.8. Expression Patterns of Induced Senescence of APXs in Maize Leaves and Internodes

The expression levels of *APXa1*, *APXb2,* and *APXc2* in pollinated and non-pollinated leaves gradually decreased from day 2 to day 24 (Figure 7). However, on day 30, the decrease in non-pollinated leaf *APXa1* expression level was more than pollinated, and non-pollinated *APXb2* and *APXc2* expression level increased. *APXa3* was depressed more in non-pollination than in pollination on day 30. The expression patterns of *APX*s in pollinated and non-pollinated internodes were similar (Figure 7). The expression of *APXa1*and *APXc2* in the two treatments decreased with the prolongation of treatment time, and *APXa2* was the opposite.

## 4. Discussion

### 4.1. Evolution of Maize APX Genes

The first *APX* gene was generated in a chloroplast matrix of unicellular green algae [9]. The five *APX* genes of four species of unicellular algae and three *APX* genes of *C. braunii* (a multicellular charophyte) are clustered in class A (Figure 1). There are two genes of *C. braunii* in class B (Figure 1). These results prove that the *APX* genes of class A is the oldest gene. Multicellular charophytes possess the *APX* genes of class B, their ancestral gene was produced by the duplication of the class A gene. The 91 genes from 12 higher plants are clustered into three groups (Figure 1). The analysis of phylogenetic tree and exon structure of *APX* genes confirm (Figure 1, Figure 2 and Figure 3, Appendix A) that higher plants have three classes of *APX* genes. The ancestral genes of class A and B of higher plants were derived from multicellular charophytes, and the ancestral gene of class C was produced by the duplication of the class B gene in the late stages of evolution (Figure 8). The analysis of genome duplication events showed that the ancestors of angiosperms first evolved into three branches; one branch is *A. trichopod*, one is the ancestor of monocots, and the other is the ancestor of dicots [25]. *A. trichopoda* did not undergo genome duplication, while the ancestors of monocots and dicots underwent genome duplication. Therefore, the *A. trichopod* genome is an invaluable reference for inferences concerning the ancestral angiosperm and subsequent genome evolution. *A. trichopoda* in each class has only one gene (Figure 1), which proves that each class of *APX* genes in angiosperms has one ancestral gene, so the ancestor of angiosperms has only three *APX* ancestral genes. The analysis of gene duplication events (Figure 4) proved that multiple genes of each class in each species were generated by genome duplication or local (or tandem) duplication of the ancestral gene (Figure 8).

The ancestor of monocotyledon underwent a WGD and subsequently evolved into the ancestors of *Musa acuminata* and other monocots approximately 100 million years ago [27]. Based on the clustering of *APX* genes of each class in *M. acuminata* and four other monocots, it can be determined whether the two copies of the ancestral *APX* formed after the first WGD were all retained in the ancestors of these species. In class A, the three genes of *M. acuminata* are clustered together with the five genes (*APXa3* branch) of the other four monocotyledons, and the eight genes of the other four monocotyledons are clustered together alone (Figure 1), meaning that one of two copies generated by WGD has been retained in the ancestor of *M. acuminata*, and two copies have both been retained in the ancestor of the other four monocotyledons. After the ancestors of *M. acuminatae* separated from the ancestors of the other four monocots, both ancestors underwent two rounds of WGDs. There was an additional WGD in maize [24,27,51,52]. Thus, one copy of the ancestral *APX* gene produced the three genes of class A of *M. acuminata* along with two rounds of WGD. Two copies of the ancestral *APX* gene produced multiple genes of the other four monocotyledons along with two or three rounds of WGD.

During the evolution of the other four monocotyledons, one of the two copies has produced the genes of *APXa1* and *APXa2* clades by two rounds of genome duplication, while the other produced the genes of the *APXa3* clade (Figure 1). The four species have only one gene in each clade, except for rice, which has two genes in the *APXa3* clade. Obviously, only one copy of the two copies of the ancestral gene of the *APXa3* clade from the first round of WGD was retained, and one copy was retained for the two copies from the second round of WGD. The two copies of the ancestral gene of *APXa1* and *APXa2* clades produced by the first round of WGD may be retained, while the two copies produced by the second round of WGD may only retain one, or the opposite may also occur. PGDD analysis proved that the genes of the *APXa1* and *APXa2* clades were generated by genome duplication events (except *OsAPX5* and *OsAPX6* in rice) (Figure 4). It is easier to find evidence of genes produced by late genome duplication events than by early genome duplication events. Therefore, we infer that the ancestral genes of the *APXa1* and *APXa2* clades only retained one of the two copies produced by the first round of WGD, while the two copies produced by the second round of WGD were all retained (Figure 9). *OsAPX5* and *OsAPX6* in rice are distributed on one chromosome and are closely linked, which proves that the two genes are produced by local (or tandem) replication. Using this method, we also carried out an evolutionary analysis of the genes of class B and C in monocots and established their evolutionary models (Appendix A).

The ancestor of dicotyledons underwent a whole genome triplication and then evolved into three ancestors in very early stages. The three ancestors were the ancestors of the following plants: (1) grape, (2) tomato and potato, and (3) many dicotyledons, such as *A. thaliana* and soybean. The ancestor of grape did not undergo a WGD event; the ancestor of tomato and potato underwent a whole genome triplication; the ancestor of *Arabidopsis thaliana* underwent two rounds of WGDs [24,26,52]. For each class of *APX* gene, based on the number of genes in grapes, the clustering of genes in dicotyledonous plants, and the analysis of gene duplication events, it is possible to determine the retention of the three copies of each class *APX* gene produced by genome triplication. In class A genes, dicotyledonous grapes have a gene that clusters together with genes from other dicotyledonous plants (Figure 1). This proves that the ancestor of a dicot plant retained only one copy after genome triplication. The two genes of Arabidopsis and tomato did not form sister clades (Figure 1). This is related to the early separation of the ancestors of Arabidopsis and tomato. The ancestors of Arabidopsis underwent two rounds of genome duplication, while the ancestors of tomato underwent genome triplication [24,52]. The two genes of Arabidopsis and tomato are produced by WGD (Figure 4). These results suggest that the ancestors of tomato retained two copies after undergoing genome triplication. The ancestors of Arabidopsis retained one of the two copies produced after the first round of genome duplication, while the two copies produced after the second round of genome duplication were all retained (Figure 9). Similarly, the evolution of Arabidopsis genes of classes B and C were analyzed, and the evolutionary models of these genes were proposed (Appendix A).

Obviously, since the ancestor of monocots was separated from the ancestors of dicots very early [41,42], the evolution pattern of maize *APX* genes is more similar to rice compared with Arabidopsis. Our research confirms that *APXa1*, *APXa2*, *APXb1*, *APXb2*, and *APXc2* of maize are orthologous genes to *OsAPX8*, *OsAPX7*, *OsAPX3*, *OsAPX4*, and *OsAPX2* of rice, respectively. *APXa3* of maize and *OsAPX5* and *OsAPX6* of rice, *APXc1.1* and *APXc1.2* of maize and *OsAPX1* of rice are orthologous genes (Figure 1). These orthologous genes may have the same or similar function. Therefore, the research results of rice genes can be applied to maize. Arabidopsis has only two *APX* genes of class A, and it is impossible to determine the orthologous relationship between these two genes and the maize genes.

### 4.2. Subcellular Localization of APX Isozymes and Differences in Tissue Expression of Maize APX Genes

Based on the analysis of the published research results [10,11,16,17,18,19,20,21,47,49] and our prediction results using four kinds of software, it can be basically determined that the APXc1.1, APXc1.2, and APXc2 isoforms in maize are located in the cytoplasm, the APXa1 isoform is located in chloroplasts, and the APXa3 isoform is located in the mitochondria. We believed that the APXb1 and APXb2 isoforms are bound to the membrane of peroxisomes, the APXa1 isoform is to bound to the thylakoid membrane of the chloroplast, and APXa2 is located in the chloroplast matrix.

The *APX* gene first appeared in the chloroplasts of unicellular green algae [9]. The gene was produced with the emergence of photosynthesis. There are three classes of *APX* genes in higher plants. Class A is older than classes B and C. Therefore, the most primitive function of class A genes is to eliminate ROS produced by photosynthesis. The maize *APXa1* gene is only highly expressed in photosynthetic organs, especially in mature leaves (Figure 5, Appendix A). The photosynthetic chain is distributed on the chloroplast thylakoid membrane, and the photosynthetic chain generates a large amount of ROS during the electron transfer process of photosynthesis [4]. Therefore, APXa1 is responsible for removing these ROS. The rice *OsAPX8* gene (the ortholog of maize *APXa1*) is only expressed in shoots [16]. OSAPX8 is a putative thylakoid-bound isoform [11,16]. The specific expression of the maize APXa1 gene only in leaves indicates that the APXa1 isoform encoded by this gene is likely to bind to the chloroplast thylakoid membrane. Unicellular green algae do not have any differentiation of roots, stems, and leaves, but angiosperms have not only a differentiation of these vegetative organs, but also a differentiation of reproductive organs. The expression site of *APXa3* is almost complementary to *APXa1*. *APXa3* is abundantly expressed in a variety of vegetative and reproductive organs at different developmental stages, except that *APXa3* is not expressed in leaves (Figure 5). *APXa2* is almost expressed in different developmental stages of various tissues (Figure 5). APXa2 is predicted to be located in the matrix of the chloroplast. *OsAPx7* (the ortholog of maize *APXa2*) accumulated preferentially in root tissues where no true chloroplasts were present [16]. Chloroplast is a plastid containing chlorophyll. Chloroplasts only exist in photosynthetic cells, but other plastids can exist in various types of cells. Teixeira et al. [16] directly replaced chloroplasts with plastids when describing the subcellular localization of APX isoforms. Based on the expression site of *APXa2*, we believe that APXa2 is more accurately located in the matrix of plastids than in the matrix of chloroplasts. APXa2 is responsible for removing ROS produced in plastids. In addition, there is a large amount of oxygen in the growth environment of terrestrial plants, so a large amount of ROS is produced by respiration. The plants have evolved the *APX* gene encoding mitochondrial isoform. The APXa3 isoform is located in the mitochondria and is responsible for the removal of ROS generated during respiration. However, the *APX* gene is not expressed in leaves, and the reasons for this phenomenon need to be further studied.

Maize *APXb1* is very low expressed in photosynthetic organs (mature leaves) and highly expressed in non-photosynthetic organs (such as roots, shoots, and immature leaves) (Figure 5, Appendix A). However, the tissue distribution of *APXb2* is complementary to that of *APXb1*, and *APXb2* is abundantly expressed in leaves and reproductive organs (Figure 5). Rice *OsAPx3* and *OsAPx4* are located in peroxisomes. The *OsAPx3* gene (the orthologous gene of maize *APXb1*) is only expressed in shoots and stems, and *OsAPx4* is expressed in flowers [16]. Peroxisomes exist in plant various cells. The *APXb1* and *APXb2* of maize are expressed in many organs, which provide further evidence for the distribution of the proteins encoded by these two genes in peroxisomes. We speculate that the *APXb1* gene retains the function of the original ancestor gene, while the *APXb2* gene appears to be subfunctional. The reasons are as follows. (1) There are class B genes in multicellular chara, and (2) multicellular chara only has the differentiation of roots, stems, and leaves, not the differentiation of reproductive organs. (3) The *APXb1* gene is mainly expressed in vegetative organs; (4) the *APXb2* gene is expressed in both vegetative and reproductive organs. Plant peroxisomes are the metabolic sites of many substances, such as photorespiration, the β-oxidation of fatty acids, the oxidation of sarcosine and xanthine, and the synthesis of indole butyric acid and jasmonic acid [53,54,55,56]. These metabolic processes produce H_2_O_2_, which can be removed by APX and distributed in the peroxisome membrane.

*APXc1.1* and *APXc2* are highly expressed in various nutritional and reproductive organs at different growth and development stages (Figure 5). Specifically, *APXc1.1* is highly expressed in roots, stems, and reproductive organs at different growth and development stages, and *APXc2* is highly expressed in vegetative organs at certain developmental stages and in almost all reproductive organs. The expression of *APXc1.2* was detected in the roots and reproductive organs but not in the aerial parts (Table 1). APXc1.1, APXc1.2, and APXc2 are located in the cytoplasm and can be responsible for removing H_2_O_2_ produced by various metabolic pathways.

### 4.3. The Difference of Maize APX Gene Expression Level in Response to Stress and Induced Senescence

Three treatments, namely infection with Ustilago maydis, drought treatment of Han21 (a maize drought-tolerant inbred line), and induced senescence, result in a decrease in the expression level of APXa1 gene in maize, while the expression level of *APXa2* increased (Figure 6A–D and Figure 7). The generation of ROS under stress conditions is caused by photorespiration, photosynthesis, mitochondrial respiration, and cell membrane NADPH (Reduced nicotinamide adenine dinucleotide phosphate) [4]. APXa1 is located in the chloroplast thylakoid membrane in the leaves and APXa2 is located in the plastid matrix of various organs (Figure 5, Table 1); these are responsible for the removal of H_2_O_2_ produced by the electron transport of the photosynthetic chain in the chloroplast and by other metabolic pathways in the plastid respectively. However, it has recently been suggested that the chloroplast is not as sensitive to ROS damage as previously thought [57]. We speculate that due to the special metabolic mechanism and structure in maize leaves, chloroplasts may not produce a large amount of ROS under stress. For example, the photosynthetic pigments of maize leaves are degraded and the light absorption capacity is reduced, resulting in a decrease in the amount of ROS produced by the photosynthetic electron transport chain. However, stress may increase the ability of chloroplasts or plastids to produce ROS through other metabolic pathways, which can lead to the accumulation of ROS in chloroplast or plastid matrix. These factors lead to the decrease of *APXa1* expression level and the increase of *APXa2* expression level under pathogen infection, drought stress, and induced senescence. During leaf senescence, the expression level of Arabidopsis *tAPX* in chloroplasts decreased [21]. This indicates that APXa1 in chloroplasts mainly plays a role under normal conditions, while APXa2 plays a role in stress treatment.

After Ustilagomaydis infection, the expression levels of *APXc1.1* and *APXc1.2* of maize leaves increase, and that of *APXb2* decreases. The expression levels of *APXc1.1* and *APXc1.2* of the leaves from Han21 (a maize drought-tolerant inbred line) remain unchanged under drought, but *APXb2* increases. The infection with pathogens causes an increase in cell membrane NADPH oxidase activity [4,58,59], which generates H_2_O_2_ that diffuses into the cytoplasm. The expression levels of *APXc1.1* and *APXc1.2* in the cytoplasm are increased (Figure 6A–C) to clear this H_2_O_2_. Under drought stress conditions, the photorespiratory pathway is also enhanced, and photorespiration is likely to account for over 70% of total H_2_O_2_ production [60]. Drought treatment reduces the water content of cells and inhibits the diffusion of H_2_O_2_ into the cytoplasm, so the expression levels of *APXc1.1* and *APXc1.2* in the cells remain unchanged. APXb2 is located in peroxisomes (Table 1) and is may be involved in eliminating H_2_O_2_ produced by photorespiration.

On the 30th day of the induction senescence treatment, the unpollinated leaves were more senescent than the pollinated leaves [36]. The expression levels of *APXb2* and *APXc2* increased in unpollinated leaves (Figure 7), which may be related to the high concentration of H_2_O_2_ produced by the recovery of membrane lipids in senescence-induced leaves.

## 5. Conclusions

In this study, the evolutionary models of *APX* genes in angiosperms are proposed for the first time, and the orthologous genes of maize and rice were determined. Maize and rice orthologous genes have similar tissue expression patterns, and the proteins encoded by these genes have similar subcellular locations. Under the prerequisite of retaining the original gene function, the excess *APX* gene copies in maize are subfunctionalized to form isoenzymes distributed in different cell compartments. In addition, maize *APX* genes expression patterns are different in various tissues and organs at different growth and development stages as well as in leaves after stress treatment and induced senescence. This may ensure that the APX isoenzymes of maize cooperate with each other to complete the elimination of H_2_O_2_.

## Figures and Tables

**Figure 1 genes-11-01204-f001:**
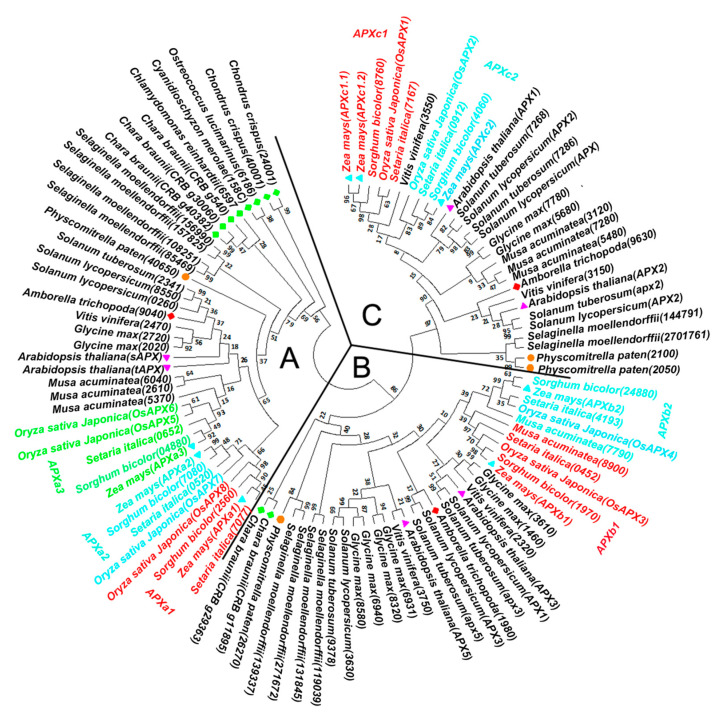
A phylogenetic tree of ascorbate peroxidase (*APX*) genes. The tree was constructed by the maximum likelihood (ML) method using MEGA6 with 1000 bootstrap replications and the Dayhoff substitution model. The gap/missing data treatment is partial deletion, and the site coverage cutoff (%) is 70. Numbers on branches indicate bootstrap support values. The protein sequences encoded by *APX* genes in the phylogenetic tree are shown in Appendix A. A, B, and C in the figure represent the gene groups of classes A, B, and C, respectively. The genes from *A. thaliana*, maize, *Amborella trichopoda*, algae, and moss are labeled by pink triangles, blue triangles, red diamonds, green diamonds, and orange dots, respectively.

**Figure 2 genes-11-01204-f002:**
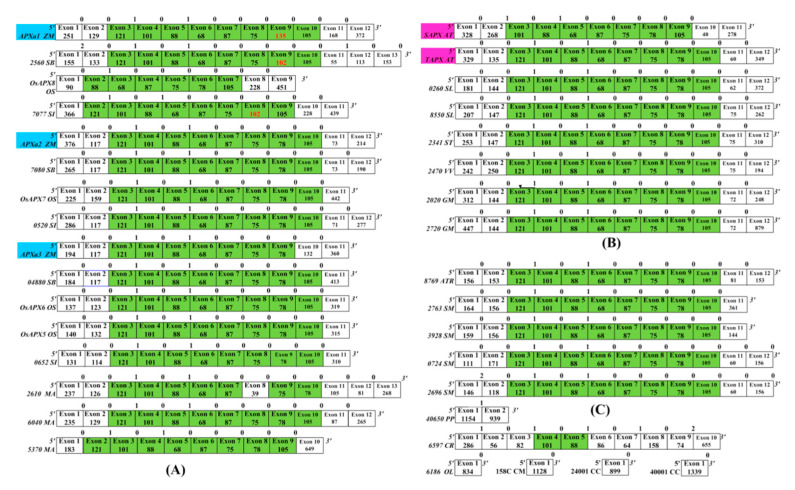
Exon/intron structures of *APX* genes of class A. (**A**) Exon/intron structures of *APX* genes of class A of monocotyledon. (**B**) Exon/intron structures of *APX* genes of class A of dicotyledon. (**C**) Exon/intron structures of *APX* genes of class A of the species except for monocotyledon and dicotyledon. The intron phases are shown on top of each exon boundary. Exon sizes are not drawn to scale. Green columns indicate highly similar exons of the same size among different species. ZM, *Zea mays*; SB, *Sorghum bicolor*; SI, *Setaria italica*; OS, *Oryza sativa Japonica*; MA, *Musa acuminata*; AT, *Arabidopsis thaliana*; GM; *Glycine max*; SL, *Solanum lycopersicum*; ST, *Solanum tuberosum*; VV, *Vitis vinifera*; ATR, *Amborella trichopoda*; SM, *Selaginella moellendorffii*; PP, *Physcomitrella patens*; CR, *Chlamydomonas reinhardtii*; OL, *Ostreococcus lucimarinus*; CM, *Cyanidioschyzon merolae*; CC, *Chondrus crispus*; CB, *Chara braunii*. The genes from *A. thaliana* and maize are labeled by pink columns and blue columns respectively.

**Figure 3 genes-11-01204-f003:**
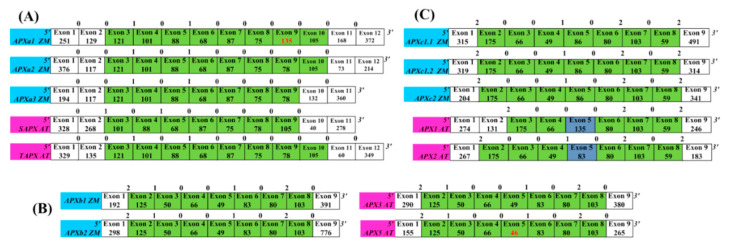
The comparison of exon/intron structures of *APX* genes in classes A, B, and C. (**A**) Exon/intron structures of *APX* genes of class A of *Z. mays* and *A. thaliana*. (**B**) Exon/intron structures of *APX* genes of class B of *Z. mays* and *A. thaliana*. (**C**) Exon/intron structures of *APX* genes of class C of *Z. mays* and *A. thaliana*. The intron phases are shown on top of each exon boundary. Exon sizes are not drawn to scale. Green columns indicate highly similar exons of the same size among different species. ZM, *Zea mays*; AT, *Arabidopsis thaliana*. The genes from *A. thaliana* and maize are labeled by pink columns and blue columns respectively.

**Figure 4 genes-11-01204-f004:**
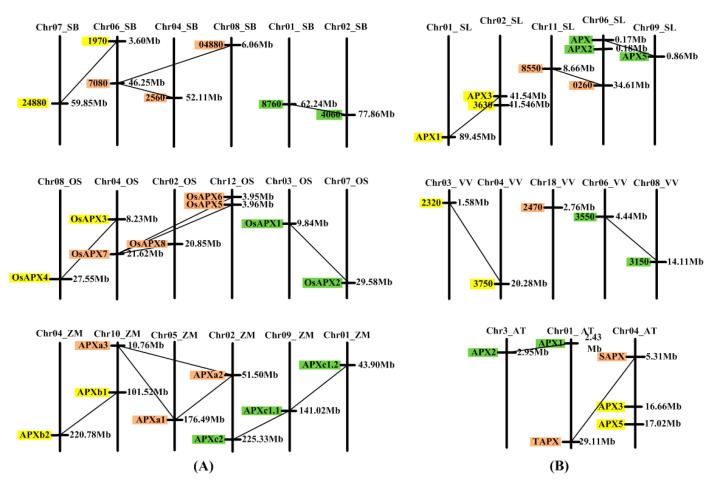
Chromosome localization of *APX* gene and whole-genome duplication (WGD) events in monocotyledon and dicotyledon. (**A**) Chromosome localization of *APX* genes and WGD events in monocotyledon. (**B**) Chromosome localization of *APX* genes and WGD events in dicotyledon. The long black bars represent chromosomes. The chromosome number and species are marked above the long black bar. For example, Chr07_SB represents chromosome 7 of sorghum. The right side of the long black bar marks the position of the gene in the chromosome, and the left side of the long black bar marks the name of the gene. The black lines between genes indicate that the genes were produced by WGD. Orange, yellow, and green indicate the genes of classes A, B, and C, respectively. ZM, *Zea mays*; SB, *Sorghum bicolor*; OS, *Oryza sativa Japonica*; T, *Arabidopsis thaliana*; SL, *Solanum lycopersicum*; VV, *Vitis vinifera*.

**Figure 5 genes-11-01204-f005:**
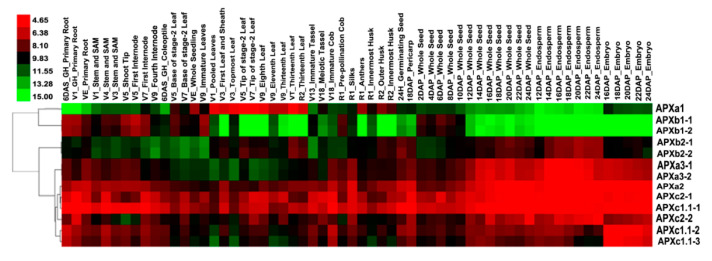
The expression patterns of *APX*s in tissues of maize at different growth and development stages. The data of expression levels of maize *APX* genes in different tissues were derived from ZM37 in PLEXdb [32]. Colous represent expression levels for each gene which are either above (red), below (green), or at the mean expression level (black). Different tissues are mentioned on top, while the genes are mentioned on the right.

**Figure 6 genes-11-01204-f006:**
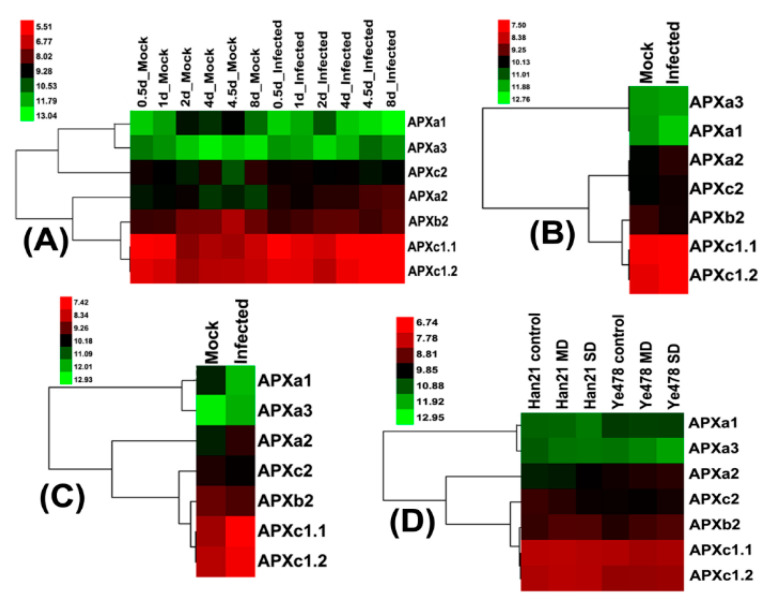
Expression patterns of maize *APX*s in response to *Ustilago maydis* infection and drought stress. (**A**–**C**): The expression patterns of maize leaf *APX*s during infection with *Ustilago maydis*. (**D**): The expression pattern of maize leaf *APX*s from two maize inbred lines under drought stress. Han21: drought-tolerant, Ye478: drought-sensitive. MD: moderate drought, SD: severe drought. The experimental data were downloaded from ZM20 (Figure 6A), ZM24 (Figure 6B), ZM38 (Figure 6C), and ZM30 (Figure 6D) in PLEXdb [32]. Colors represent expression levels for each gene, which are either above (red), below (green), or at the mean expression level (black). Different treatments are mentioned on top, while the genes are mentioned on the right.

**Figure 7 genes-11-01204-f007:**
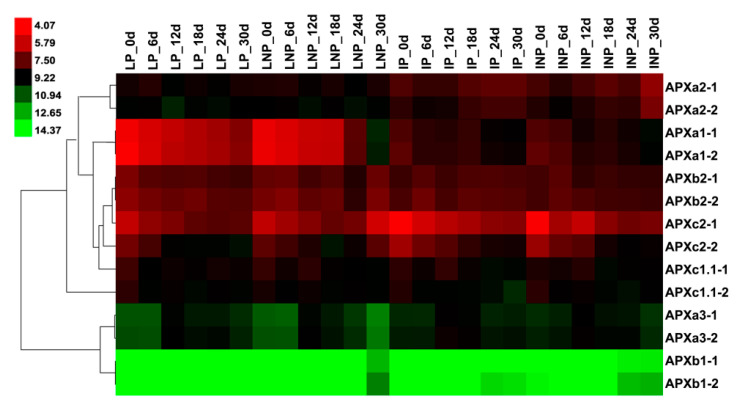
Expression patterns of *APX* genes in maize leaves and internodes under induced senescence. The experimental data were downloaded from ZM54 in PLEXdb [32]. Colors represent the expression levels for each gene, which are either above (red), below (green), or at the mean expression level (black). Developmental stages are mentioned on top, while the genes are mentioned on the right.

**Figure 8 genes-11-01204-f008:**
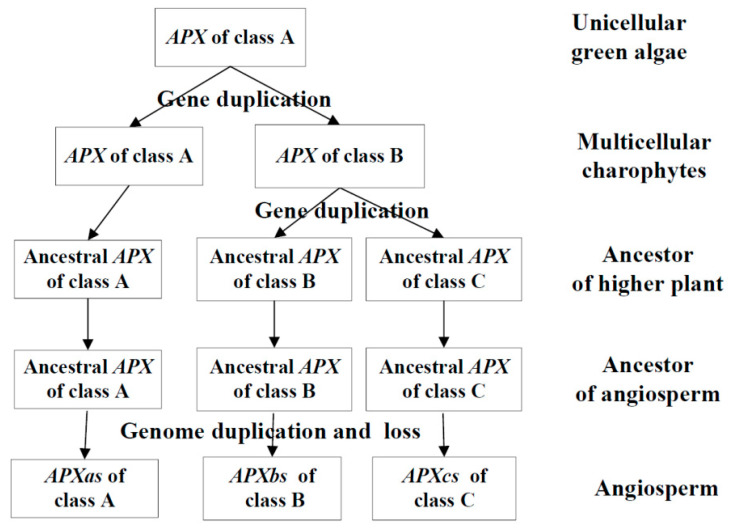
The origin of the angiosperm *APX* genes. Plant species are mentioned on the right.

**Figure 9 genes-11-01204-f009:**
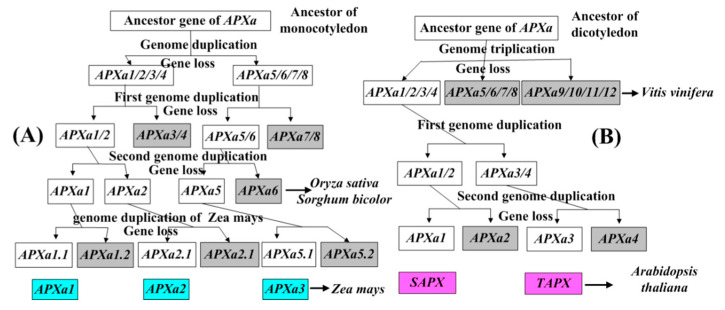
A model for the evolution of the angiosperm *APX* genes of class A. (**A**) A model for the evolution of monocotyledon. (**B**) A model for the evolution of dicotyledon. Gray columns indicate missing genes. The genes from *A. thaliana* and maize are labeled by pink columns and blue columns, respectively.

**Table 1 genes-11-01204-t001:** Predicted subcellular localization of maize *APX* genes.

Gene Name	KnowPredsite II	Target P 1.1	CELLO v.2.5	WoLF PSORT	Result
*APXa1*	Plastid (100) Chloroplast (100) Plasma membrane (40) Mitochondrion (20)	Chloroplast (1)	Chloroplast (2.2)	Chloroplast (13)	Thylakoid membrane of the chloroplast
*APXa2*	Plastid (100) Chloroplast (100) Plasma membrane (40) Mitochondrion (20)	Chloroplast (4)	Mitochondrion (2.3) Chloroplast (1.4)	Chloroplast (11) Mitochondrion (3)	Matrix of chloroplast (matrix of plastid)
*APXa3*	Plastid (100) Chloroplast (100) Plasma membrane (40) Mitochondrion (20)	Mitochondrion (1)	Mitochondrion (2.6) Chloroplast (1.2)	Chloroplast (9) Mitochondrion (5)	Mitochondrion
*APXb1*	Plasma membrane (100) Peroxisome (40)	Other (3)	Cytoplasm (2.8) Mitochondrion (0.9)	Cytoplasm (7.5)	Peroxisome
*APXb2*	Plasma membrane (100) Peroxisome (40)	Mitochondrion (5) Other (1)	Cytoplasm (1.9) Mitochondrion (1.2) Chloroplast (1.0)	Chloroplast (13)	Peroxisome
*APXc1.1*	Cytoplasm (100)	Other (5)	Cytoplasm (2.4)	Mitochondrion (6) Cytoplasm (4)	Cytoplasm
*APXc1.2*	Cytoplasm (100)	Other (5)	Cytoplasm (2.6)	Cytoplasm (4) Chloroplast (3)	Cytoplasm
*APXc2*	Cytoplasm (100)	Other (3)	Cytoplasm (2.6)	Cytoplasm (4)	Cytoplasm

In the table, the numbers in parentheses are the correct answer (KnowPredsite II), reliability class (Target P 1.1), reliability (CELLO v.2.5). Larger numbers have more predictive reliability.

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
