# Peer review of "Molecular Evolution of Maize Ascorbate Peroxidase Genes and Their Functional Divergence"

_genes, 2020, doi:10.3390/genes11101204_

Round 1

Reviewer 1 Report

In this paper, authors studied the Ascorbate peroxidase (APX) gene family and evolutionary relation and functional divergence in maize APX gene through bioinformatics approaches.

Major comments

The manuscript is still very rough. The authors need to rearrange or rewrite extensively on the discussion part. Since evolution is a complicated phenomenon, authors were too quick on concluding how APX genes evolved from ancestral genes from other species. And, that also with the phylogenic tree of limited species, prediction localization, and only selected event (only one) each of biotic and abiotic expression data from the internet.

The authors retrieved the APX protein sequences of 18 species for the analysis of phylogeny, splice site, and duplication events of APX genes but fail to explain on what basis those species were selected.

Since Ozyigit et al 2016 and Liu et al 2012 also performed a localization of Zm APXs, authors need to give a reason on what novel approach they used and how the prediction of localization weighs regarding evolutionary and functional divergence parameter.

Authors have drawn the conclusion on the evolutionary relationship of angiosperm (maize) APX based on the phylogenic tree (with a limited number of other plant species). However, it is a complicated phenomenon. A more extensive study including a lot more plant species could be a solution for that.

Specific comments

Most of the figures are very vague to understand. Authors need to explain the figure more or figure should not be the only figure title.

Page2, line 57- authors mentioned that the reason for in-depth comparative genomics of maize they are doing is that their preliminary analysis revealed there were 11 APX genes in maize, but they never mentioned in result/discussion or did not put that analysis. It would have been different than what other researchers found ( all other found 8 APX in maize). So why authors proceed with only 8 APX and any reason behind that?

Table 1, As the predicted results of the 4 predictions for localization were not consistent, authors need to explain the basis for the final prediction results. On the note, the authors mentioned that a larger number in parenthesis has more predictive reliability. However, that seems not what authors put in the result column. For example, APXa2-KnowPredsite II [plastid(100), chloroplast(100), plasma membrane(40), mitochondrion(20)]; TargetP1.1[chloroplast(4)]; CELLO v2.5[mitochondrion(2.3), chloroplast(1.4)]; WoLF PSORT[chloroplast(11), mitochondrion(3)]. So, it is a bit confusing about how mitochondrion is a final result? The authors need to explain in detail.

Figure 2, authors could have labeled three sub-figures as A, B, C instead of writing monocotyledons, dicotyledon at the top, and put figure note.

Author Response

Major comments

The manuscript is still very rough. The authors need to rearrange or rewrite extensively on the discussion part. Since evolution is a complicated phenomenon, authors were too quick on concluding how APX genes evolved from ancestral genes from other species. And, that also with the phylogenic tree of limited species, prediction localization, and only selected event (only one) each of biotic and abiotic expression data from the internet.

[Qu] The manuscript has been extensively rewritten. In the “Results” and “Discussion” sections, the basis for establishing the APX gene evolution model has been explained in detail now. There is not much data in PLEXdb about the gene expression research of maize under stress treatment. At the same time, considering the length of the manuscript, no more experimental data was selected.

The authors retrieved the APX protein sequences of 18 species for the analysis of phylogeny, splice site, and duplication events of APX genes but fail to explain on what basis those species were selected.

[Qu] In the “Materials and Methods” section, the reasons for the selection of species for systematic evolution analysis has been explained now.

Since Ozyigit et al 2016 and Liu et al 2012 also performed a localization of Zm APXs, authors need to give a reason on what novel approach they used and how the prediction of localization weighs regarding evolutionary and functional divergence parameter.

[Qu]This has been explained in the “Result” section.

Authors have drawn the conclusion on the evolutionary relationship of angiosperm (maize) APX based on the phylogenic tree (with a limited number of other plant species). However, it is a complicated phenomenon. A more extensive study including a lot more plant species could be a solution for that.

[Qu] By referring to the existing research results of angiosperm genome replication and selecting species with characteristics of genome replication as a reference, we established an angiosperm (maize) APX evolutionary relationship model. For details, please check the “Introduction”, “Materials and Methods”, and “Results” sections in the manuscript.

Specific comments

Most of the figures are very vague to understand. Authors need to explain the figure more or figure should not be the only figure title.

[Qu] The manuscript has been extensively revised. Several figures are explained in detail.

Page2, line 57- authors mentioned that the reason for in-depth comparative genomics of maize they are doing is that their preliminary analysis revealed there were 11 APX genes in maize, but they never mentioned in result/discussion or did not put that analysis. It would have been different than what other researchers found ( all other found 8 APX in maize). So why authors proceed with only 8 APX and any reason behind that?

[Qu] This part has been revised in the manuscript.

Table 1, As the predicted results of the 4 predictions for localization were not consistent, authors need to explain the basis for the final prediction results. On the note, the authors mentioned that a larger number in parenthesis has more predictive reliability. However, that seems not what authors put in the result column. For example, APXa2-KnowPredsite II [plastid(100), chloroplast(100), plasma membrane(40), mitochondrion(20)]; TargetP1.1[chloroplast(4)]; CELLO v2.5[mitochondrion(2.3), chloroplast(1.4)]; WoLF PSORT[chloroplast(11), mitochondrion(3)]. So, it is a bit confusing about how mitochondrion is a final result? The authors need to explain in detail.

[Qu] It has been elaborated in the “Result” and “Discussion” sections now.

Figure 2, authors could have labeled three sub-figures as A, B, C instead of writing monocotyledons, dicotyledon at the top, and put figure note.

[Qu] It has been revised as required.

Submission Date

08 September 2020

Date of this review

18 Sep 2020 21:19:42

Reviewer 2 Report

Qu et al analyse the APX gene family in Zea mays based on publicly available data. While they demonstrate very nicely the potential of reusing public datasets, there are several technical issues that need to be addressed to make this study technical solid (see comments below).

Additionally, I see one major issue with the conclusions drawn from the results and the way these are presented. The authors imply (and actually write in some places) in the discussion section that the analysed genes have certain functions in Zea mays. It is valid to generate hypotheses about enzyme functions based on the demonstrated functions of orthologs. However, it is extremely important to make it very clear throughout the manuscript that only hypotheses are presented. Not a single APX enzyme function of Zea mays was actually shown.

Another major issue is the lack of novel/surprising findings. Since the authors assume that all functions in Zea mays are identical to rice, this is not surprising. I think the inclusion of additional species might help to identify something.

Major comments:

1) There seems to be a confusion about some APX genes in rice (line38 - 39). If four belong to the plastids, they cannot be the only mitochondria ones at the same time. The authors might want to check this and rephrase the sentence if appropriate.

2) The authors state in their introduction that this study is a preliminary analysis. In this case, the authors should perform the actual analysis and re-submit their final results for publication. If the current analysis is the final one, the sentences should be rephrased.

3) It is not clear how the sequences in various species were identified. Was this just a selection based on available annotation? A full list of all sequences used in this analysis should be included in the supplements. Ideally, all coding sequences would be included as FASTA files.

4) Using an e-value of 10 as cutoff will return many unspecific hits. I would expect to see many more sequences in the phylogenetic tree. Were additional filter criteria applied? The authors might want to explain their choice of BLAST parameters. Also, which version of BLAST was used?

5) All phylogenetic analyses should be repeated on the nucleotide level using a codon alignment. Also, the authors might want to check if the tree topology changes substantially if the coverage cutoff is reduced to 10%. As the conclusions rely heavily on the correctness of the tree topology, the authors might want to try different tools for the tree construction (e.g. RAxML-NG).

6) It appears that the authors analysed peptide sequences, but are writing about genes in the next paragraph. Consistent terminology should be used clearly throughout the manuscript. Genes might give rise to different transcripts/peptides. How were these isoforms handled in this study?

7) Fig1: The authors should show full species names in tree. It is complicated to understand the tree if all abbreviations need to be looked up in the legend.

8) What is about the splice sites? Do all genes have the canonical GT-AG motif or are there any non-canonical splice sites conserved in this gene family? If the exon-intron structure is highly conserved, there might be some conservation inside the introns too.

9) The authors are frequently using "replication", but they probably should use "duplication" (e.g. line 191). Why should replication time effect genome duplications?

10) How is the chromosomal distribution similar (line198)? The chromosome numbers differ between plant species. "Homologous" chromosomes in different species might not have matching numbers. The legend of Fig. 4 needs to explain what the different abbreviations mean and which chromosomes are displayed.

11) The authors performed only predictions of the subcellular localization (line210-line225). This needs to be clearly indicated as a prediction throughout the text.

12) What is the conclusion from the gene expression analysis? Is there a general picture? The font size in Fig. 5 appears very small and is not readable.

13) Fig.5/6/7: The samples should be labelled with clear names to facilitate an easy interpretation. Why are some genes not displayed in Fig.6? What is the purpose of the hierarchical clustering of gene expression? How does the clustering by gene expression compare to the phylogenetic relationship?

14) The author speculate about gene loss. Have the authors checked that there are no additional (pseudo)genes in any of the Zea mays genomes? It is possible that some gene copies are lost in specific cultivars, but are still present in others.

15) Orthologous genes are likely to have a similar function (line 372). However, it cannot be assumed that all orthologs will have the very same function. Especially when transferring functional annotations across a long phylogenetic distance, there is a substantial uncertainty involved. The authors might want to rephrase their sentence to better indicate this. Also, I strongly disagree with the statement that the inference of orthologs to A. thaliana would be impossible. The authors need to include additional species to cover the large phylogenetic gap. If the identification of orthologs is not feasible, it would be possible to identify at least orthogroups.

16) The authors should rephrase their discussion (line 376-line 394 and also the following paragraphs) to clearly indicate that these are assumptions derived from research on rice orthologs. It is important to clearly indicate that the function was not shown in Zea mays yet. This study only contains a phylogenetic analysis and expression analyses. However,t he actually function of the enzymes was not shown. How can the authors exclude that some of these genes are encoding non-functional products?

17) If I understood it correctly, the discussion is mainly a summary about APX findings in rice (and other species). Therefore, it is important to accurately reflect this in the headings. Currently, the authors imply that APX was characterized in Zea mays, but all presented findings are from different species.

18) The conclusion should be written more concisely and should only present key/novel findings.

19) What is about divergence of sequences? Is it possible to see the selection pressure on different sequences? Is there any sign of sub-/neofunctionalisation after duplication (except for different expression domains)? Are there diagnostic residues which might hint towards changes of key enzyme properties (e.g. substrate affinity)?

Minor comments:

line 10-11: one enzyme encoded by multiple genes?

line 12: "predictive software" is not informative and should be replaced

line 46: Arabidopsis thaliana could be shortened to "A. thaliana" and not "Arabidopsis" after initial introduction. There are also other Arabidopsis species.

line 55: researches > studies

124: local replication > tandem duplication?

line 146 (and elsewhere): Musa acuminate > Musa acuminata (common issue caused by autocorrection)

line 165: base on > on the basis of

line 186: replication > duplication

line320-322: The authors might want to rephrase the sentence of the 3 ancestors. There is usually just one ancestor per evolutionary lineage.

line 330: has > have

Fig. 9: The authors might want to increase the font size.

line 344: A model is always hypothetical. The authors might want to call it just "model".

line346: The authors might want to check this "APXb2 and APXb2".

line 353: The authors might want to rephrase "WGD replication". The intended meaning is probably just "WGD (whole genome duplication)" which already includes the duplication.

line 423 (and elsewhere): "Ustilagomaydis" is not a condition on its own. The authors might want to rephrase some sentences

This is an incomplete list of minor issues. The manuscript would benefit from careful proof reading.

Author Response

Major comments:

1) There seems to be a confusion about some APX genes in rice (line38 - 39). If four belong to the plastids, they cannot be the only mitochondria ones at the same time. The authors might want to check this and rephrase the sentence if appropriate.

[Qu] It has been revised.

2) The authors state in their introduction that this study is a preliminary analysis. In this case, the authors should perform the actual analysis and re-submit their final results for publication. If the current analysis is the final one, the sentences should be rephrased.

[Qu] The “Introduction” section has been revised.

3) It is not clear how the sequences in various species were identified. Was this just a selection based on available annotation? A full list of all sequences used in this analysis should be included in the supplements. Ideally, all coding sequences would be included as FASTA files.

[Qu] It has been explained in detail in the section “Materials and Methods”. The protein sequence encoded by the gene has been uploaded as Table S2.

4) Using an e-value of 10 as cutoff will return many unspecific hits. I would expect to see many more sequences in the phylogenetic tree. Were additional filter criteria applied? The authors might want to explain their choice of BLAST parameters. Also, which version of BLAST was used?

[Qu] It has been explained in detail in the “Materials and Methods” section.

5) All phylogenetic analyses should be repeated on the nucleotide level using a codon alignment. Also, the authors might want to check if the tree topology changes substantially if the coverage cutoff is reduced to 10%. As the conclusions rely heavily on the correctness of the tree topology, the authors might want to try different tools for the tree construction (e.g. RAxML-NG).

[Qu] Phylogenetic analysis generally chooses to use protein sequences. The evolutionary analysis of genes is not only limited to the phylogenetic tree, but also verifies the results of the phylogenetic tree through the splicing site of the gene, gene duplication and chromosome distribution of the gene. These have been explained in detail in the “Results” section of the article.

6) It appears that the authors analysed peptide sequences, but are writing about genes in the next paragraph. Consistent terminology should be used clearly throughout the manuscript. Genes might give rise to different transcripts/peptides. How were these isoforms handled in this study?

[Qu] It has been revised.

7) Fig1: The authors should show full species names in tree. It is complicated to understand the tree if all abbreviations need to be looked up in the legend.

[Qu] Figure 1 has been revised as required.

8) What is about the splice sites? Do all genes have the canonical GT-AG motif or are there any non-canonical splice sites conserved in this gene family? If the exon-intron structure is highly conserved, there might be some conservation inside the introns too.

[Qu] The “Results” and “Discussion” of the manuscript are explained these in detail.

9) The authors are frequently using "replication", but they probably should use "duplication" (e.g. line 191). Why should replication time effect genome duplications?

[Qu] It has been corrected "replication" and "duplication" in the manuscript.

10) How is the chromosomal distribution similar (line198)? The chromosome numbers differ between plant species. "Homologous" chromosomes in different species might not have matching numbers. The legend of Fig. 4 needs to explain what the different abbreviations mean and which chromosomes are displayed.

[Qu] The “Results” and “Discussion” of the manuscript are explained these in detail.

11) The authors performed only predictions of the subcellular localization (line210-line225). This needs to be clearly indicated as a prediction throughout the text.

[Qu] It has been revised.

12) What is the conclusion from the gene expression analysis? Is there a general picture? The font size in Fig. 5 appears very small and is not readable.

[Qu] In the “Discussion” section, it has been explained the conclusions for gene expression analysis. The font in Figure 5 has been enlarged.

13) Fig.5/6/7: The samples should be labelled with clear names to facilitate an easy interpretation. Why are some genes not displayed in Fig.6? What is the purpose of the hierarchical clustering of gene expression? How does the clustering by gene expression compare to the phylogenetic relationship?

[Qu] Figure 5/6/7 has been re-annotated. Since these genes are not in the maize microarray used in the experiment, these genes are not shown in Figure 6. Figures 5 and 7 are in a similar situation.

14) The author speculate about gene loss. Have the authors checked that there are no additional (pseudo)genes in any of the Zea mays genomes? It is possible that some gene copies are lost in specific cultivars, but are still present in others.

[Qu] The method for determining APX gene in maize has been explained in detail in the “Results” section of the manuscript.

15) Orthologous genes are likely to have a similar function (line 372). However, it cannot be assumed that all orthologs will have the very same function. Especially when transferring functional annotations across a long phylogenetic distance, there is a substantial uncertainty involved. The authors might want to rephrase their sentence to better indicate this. Also, I strongly disagree with the statement that the inference of orthologs to A. thaliana would be impossible. The authors need to include additional species to cover the large phylogenetic gap. If the identification of orthologs is not feasible, it would be possible to identify at least orthogroups.

[Qu] It has been revised.

16) The authors should rephrase their discussion (line 376-line 394 and also the following paragraphs) to clearly indicate that these are assumptions derived from research on rice orthologs. It is important to clearly indicate that the function was not shown in Zea mays yet. This study only contains a phylogenetic analysis and expression analyses. However,t he actually function of the enzymes was not shown. How can the authors exclude that some of these genes are encoding non-functional products?

[Qu] It has been revised.

17) If I understood it correctly, the discussion is mainly a summary about APX findings in rice (and other species). Therefore, it is important to accurately reflect this in the headings. Currently, the authors imply that APX was characterized in Zea mays, but all presented findings are from different species.

[Qu] The subcellular location of APX isozymes mainly refers to the research results of rice. However, the expression of the eight APX genes of maize at different developmental stages of maize and the changes in gene expression levels under stress conditions have proved that the mechanisms of expression regulation of these genes are different. This causes a difference in functionality.

18) The conclusion should be written more concisely and should only present key/novel findings.

[Qu] The conclusion has been modified.

19) What is about divergence of sequences? Is it possible to see the selection pressure on different sequences? Is there any sign of sub-/neofunctionalisation after duplication (except for different expression domains)? Are there diagnostic residues which might hint towards changes of key enzyme properties (e.g. substrate affinity)?

[Qu] This manuscript did not study the selection pressure of the APX gene in maize, nor did it analyze the protein structure encoded by the APX gene. The differences in subcellular localization and gene expression patterns indicate that the newly generated genes have undergone subfunctional differentiation.

Minor comments:

line 10-11: one enzyme encoded by multiple genes?

[Qu] It has been revised.

line 12: "predictive software" is not informative and should be replaced

line 46: Arabidopsis thaliana could be shortened to "A. thaliana" and not "Arabidopsis" after initial introduction. There are also other Arabidopsis species.

[Qu] It has been revised.

line 55: researches > studies

[Qu] It has been revised.

124: local replication > tandem duplication?

[Qu] It has been revised.

line 146 (and elsewhere): Musa acuminate > Musa acuminata (common issue caused by autocorrection)

[Qu] It has been revised.

line 165: base on > on the basis of

[Qu] It has been revised.

line 186: replication > duplication

[Qu] It has been revised.

line320-322: The authors might want to rephrase the sentence of the 3 ancestors. There is usually just one ancestor per evolutionary lineage.

[Qu] Yes. It has been revised.

line 330: has > have

[Qu] It has been revised.

Fig. 9: The authors might want to increase the font size.

[Qu] It has been revised.

line 344: A model is always hypothetical. The authors might want to call it just "model".

[Qu] It has been revised.

line346: The authors might want to check this "APXb2 and APXb2".

[Qu] It has been revised.

line 353: The authors might want to rephrase "WGD replication". The intended meaning is probably just "WGD (whole genome duplication)" which already includes the duplication.

[Qu] It has been revised.

line 423 (and elsewhere): "Ustilagomaydis" is not a condition on its own. The authors might want to rephrase some sentences

[Qu] It has been revised.

This is an incomplete list of minor issues. The manuscript would benefit from careful proof reading.

Submission Date

08 September 2020

Date of this review

11 Sep 2020 10:53:23

Round 2

Reviewer 1 Report

The revised manuscript entitled “Molecular evolution of maize ascorbate peroxidase genes and their functional divergence” looks a lot better. The authors did a good job revising extensively especially figure notes and discussion part. I found some typos or might be mistakes while editing. For example, in line 715, instead of “Teixeira found that…” it should have been "Teixeira et al (2006) found that..." Also, there are other grammatical errors in some instances. On my random checking, I found reference no. 35 to be not in reference format (journal name should be in abbreviation)

I strongly recommend authors to go over the manuscript multiple times and correct all the typos and grammatical mistakes. Also, revise all the references to the correct format.

Author Response

Dear reviewer,

Thank you very much for your valuable and constructive comments on this manuscript.

There are a lot of contents need to be modified in the first time, and the time for modification is limited. The revised manuscript was not fully checked and polished. I also did not reply to each of your comments in detail and did not provide the modified line number. I apologize for causing trouble to your review!

The manuscript was revised according to your comments. The sentence errors, the format and the links of the references in the manuscript have been all revised. In order to establish hyperlinks, this revision deletes the references in the first manuscript.

For the second revision, please refer to the revisions made by the revisionist named "Revision 2" in the manuscript. The other revision is the first revision.

Best regards,

Chunxiang Qu

Let me start to answer your comments:

The revised manuscript entitled “Molecular evolution of maize ascorbate peroxidase genes and their functional divergence” looks a lot better. The authors did a good job revising extensively especially figure notes and discussion part. I found some typos or might be mistakes while editing. For example, in line 715, instead of “Teixeira found that…” it should have been "Teixeira et al (2006) found that..." Also, there are other grammatical errors in some instances. On my random checking, I found reference no. 35 to be not in reference format (journal name should be in abbreviation)

I strongly recommend authors to go over the manuscript multiple times and correct all the typos and grammatical mistakes. Also, revise all the references to the correct format.

  1. We have revised the format of all cited references in the manuscript (lines 46, 48, 49, 53, 144, 249, etc.).
  2. We corrected typos and grammatical errors in the manuscript (lines 23, 74, 102, 103, 142, etc.).
  3. We corrected the incorrect format in the references one by one (lines 1001-1138).

Reviewer 2 Report

The authors addressed some of my comments, but do not provide line numbers to easily check changes. Also, many comments have not been addressed. Large sections of this manuscript have been rewritten and generally improved the quality. However, there are still several issues.

1/2) This looks like a whole new introduction.

a) line 49: Is it possible that these research groups were working on different varieties of these species? It would not be surprising to see small differences between varieties. How many genes are there in Arabidopsis? An integrated phylogenetic analysis of the 6 and 8 reported sequences should tell this.

b) The sentence in line 71 needs to be rephrased. "finalized" is probably not what the authors intend to say here.

c) line 78 - line 79: The genomes are complex, but it is possible to study their evolution. This sentence should be removed, because it does not provide any details.

d) line 92 - line 93: Are there any benchmarking studies for subcellular localization prediction tools?

e) The sentence in line 96 should be removed, because it is very superficial.

3) line 136 - line 140 are introduction not methods. The sentence in line 141 needs to be rephrased. The authors still not explain HOW the search for candidate genes was performed. Which tool was used? Which parameters? Unfortunately, I cannot access the supplementary files. The files seems to be broken.

4) I cannot find the explanation for "Using an e-value of 10 as cutoff will return many unspecific hits. I would expect to see many more sequences in the phylogenetic tree. Were additional filter criteria applied? The authors might want to explain their choice of BLAST parameters. Also, which version of BLAST was used?". The authors should provide line numbers to make this easy to check.

5) No, the nucleotide sequences has more signal. Aligning peptide sequences and translating into the nucleotide sequences can improve the resulting phylogenetic tree.

6) Where was this addressed "It appears that the authors analysed peptide sequences, but are writing about genes in the next paragraph. Consistent terminology should be used clearly throughout the manuscript. Genes might give rise to different transcripts/peptides. How were these isoforms handled in this study?"?

7) ok

8) What is about the splice sites? Do all genes have the canonical GT-AG motif or are there any non-canonical splice sites conserved in this gene family? If the exon-intron structure is highly conserved, there might be some conservation inside the introns too.

9) The authors are frequently using "replication", but they probably should use "duplication" (e.g. line 390). Why should replication time effect genome duplications?

10) How is the chromosomal distribution similar (line198)? The chromosome numbers differ between plant species. "Homologous" chromosomes in different species might not have matching numbers. The legend of Fig. 4 needs to explain what the different abbreviations mean and which chromosomes are displayed.

11) The authors performed only predictions of the subcellular localization. This needs to be clearly indicated as a prediction throughout the text (e.g. header of Table1).

12) What is the conclusion from the gene expression analysis? Is there a general picture? The font size in Fig. 5 appears very small and is not readable.

13) If candidate genes are missing from the microarray, the authors should use RNAseq data sets instead which are not restricted to certain gene sets. How does the clustering by gene expression compare to the phylogenetic relationship?

14) The author speculate about gene loss. Have the authors checked that there are no additional (pseudo)genes in any of the Zea mays genomes? It is possible that some gene copies are lost in specific cultivars, but are still present in others.

15) Orthologous genes are likely to have a similar function (line 372). However, it cannot be assumed that all orthologs will have the very same function. Especially when transferring functional annotations across a long phylogenetic distance, there is a substantial uncertainty involved. The authors might want to rephrase their sentence to better indicate this. Also, I strongly disagree with the statement that the inference of orthologs to A. thaliana would be impossible. The authors need to include additional species to cover the large phylogenetic gap. If the identification of orthologs is not feasible, it would be possible to identify at least orthogroups.

16)

a) The authors should rephrase their discussion (originally line 376-line 394 and also the following paragraphs) to clearly indicate that these are assumptions derived from research on rice orthologs. It is important to clearly indicate that the function was not shown in Zea mays yet. This study only contains a phylogenetic analysis and expression analyses. However,t he actually function of the enzymes was not shown. How can the authors exclude that some of these genes are encoding non-functional products?

b) line 565-566: "These results prove that the APX genes of class A are the oldest gene." There could be an older lineage which is absent from the investigated species. Therefore, conclusions should be phrased more carefully.

17/18)

a) "Maize and rice orthologous genes have similar tissue expression patterns" This sentence in the conclusion appears to be conflicting with the authors' response "However, the expression of the eight APX genes of maize at different developmental stages of maize and the changes in gene expression levels under stress conditions have proved that the mechanisms of expression regulation of these genes are different."

b) The conclusion is inaccurate, because previous studies already addressed the APX evolution e.g. https://link.springer.com/article/10.1007/s00438-017-1413-2 and https://www.sciencedirect.com/science/article/pii/S0888754320300379. It appears that some relevant references might be missing here.

19) The main question that I see is "why are there so many gene copies?". The authors show different expression domains which could be the explanation. However, to conclude that this is the main reason, it would be necessary to check for other changes.

20) line 304-line307 should be moved to the discussion. Although these questions cannot be completely answered, I think they could be addressed. It might not be possible to provide perfect answer, but it would be possible to predict e.g. the number of gene copies in an ancestor. I can see from the tree that there were at least 3 copies.

21) The method section generates the impression that these experiments would have been part of this study. If the experiments are described in the original studies, it should be sufficient to just cite them.

22) line 314-line319: How can the authors tell that the ancestral gene has not lost its introns in algae species?

23) What are "intron phases" (line 336)? Is this supposed to be the phase of coding sequences in the exons?

Minor comments:

line 329: Musa acuminate > Musa acuminata

line 191 (and others): "cDNA sequences" > transcript sequences

line 314 and line 331: Physcomitrella paten > Physcomitrella patens

This is an incomplete list. The manuscript would benefit from careful proofreading and language editing.

Author Response

Dear reviewer,

Thank you very much for your valuable and constructive comments of this manuscript.

There are a lot of contents need to be modified in the first time, and the time for modification is limited. Therefore, I did not reply to each of your comments in detail and did not provide the modified line number. I apologize for causing trouble to your review!

Before answering your comments and suggestions in detail, I would like to explain the innovative points of this manuscript.

The main innovation is that an evolutionary model of APX genes in angiosperms is established for the first time in this manuscript. We have conducted evolutionary studies on the cryptochrome gene in zebrafish and established an evolutionary model of this gene (Liu C, Hu J, Qu C, Wang L, Huang G, Niu P, et al. Molecular evolution and functional divergence of zebrafish (Danio rerio) cryptochrome genes. Sci Rep-uk. 2015;5:8113). Many references, including 2 articles provided by you (17/18) b)), have conducted genome-wide analysis on multiple genes in angiosperms. However, compared with animal genome duplication, plant genome duplication is much more complicated, so no reports have been found to establish angiosperm gene evolution models. This manuscript takes the APX gene in plants as the research object. Through the research, we have established a variety of angiosperm APX gene evolution models, and clarified the orthologous genes of rice and maize. This lays the foundation for applying the research results of rice and other monocotyledonous species to the study of the APX gene function of maize. This manuscript focuses on the study of the evolutionary history of maize APX genes, and has not fully studied the functional differences of maize APX genes.

The manuscript was revised according to your comments. The sentence errors, the format and the links of the references in the manuscript have been all revised. In order to establish hyperlinks, this revision deletes the references in the first manuscript.

For the second revision, please refer to the revision made by the revisionist named "Revision 2" in the manuscript. The other revisionist was made the first revision.

Best regards,

Chunxiang Qu

Let me start to answer your comments:

1/2) This looks like a whole new introduction.

  1. a) line 49: Is it possible that these research groups were working on different varieties of these species? It would not be surprising to see small differences between varieties. How many genes are there in Arabidopsis? An integrated phylogenetic analysis of the 6 and 8 reported sequences should tell this.

These research groups did not study the small differences between varieties. In the reports that we have found on genome-wide analysis of genes, we have not seen any research on the analysis of differences between different varieties. Upon seeing your question, we also became interested in the number of APX genes in different varieties. The rice variety used by Teixeira et al. [11] is “Oryza sativa L. var. indica cv. Taim 7”. “Oryza sativa Japonica” is used in this study. There is also a variety in e! Ensembl Plants called “Oryza sativa Indica". In this manuscript, although our research variety is different from the research variety of Teixeira et al. [11], the types and numbers of APX genes are the same. Before responding to your comments, we identified the APX genes in "Oryza sativa Indica" and found that it was consistent with the first two varieties. This indicates that there may be no difference in the class and number of APX genes among different varieties with close genetic relationships.

We have performed APX identification on plants of the same genus but of different species. There are 3 kinds of plants in e! Ensembl Plants, namely Arabidopsis thaliana, Arabidopsis lyrata and Arabidopsis halleri. A. lyrata differentiated from the original 8-ploid A. thaliana 10 million years ago. Its genome is large (206.7Mb), but modern A. thaliana has lost a large number of chromosomal fragments and has a smaller genome (125Mb). We also identified the APX genes of these two plants. It was found that each of the three species has an orthologous gene of APX4 and APX6. These genes are not true APX genes, because they have mutations in key sites of protein enzyme activity. A. thaliana has 6 truly active APX genes, and 8 APX genes have been identified in both A. lyrata and A. halleri.

In A. lyrata, there are 2 genes of class A, 4 genes of class B, and 2 genes of class C. In A. halleri, there are 2 genes of class A, 5 genes of class B, and one gene of class C. Obviously, the numbers of APX genes in these two species are different from A. thaliana, but the classes of genes are the same.

It can be seen that there is little difference in the number of genes within species, but there are differences between species. Therefore, I revised "Arabidopsis, rice, and maize" in line 47 of the manuscript to "Arabidopsis thaliana, Oryza sativa Japonica and Zea mays", which is more accurate. Thanks for your reminding.

  1. b) The sentence in line 71 needs to be rephrased. "finalized" is probably not what the authors intend to say here.

For the revision, please check line 73 of the manuscript.

  1. c) line 78 - line 79: The genomes are complex, but it is possible to study their evolution. This sentence should be removed, because it does not provide any details.

Compared with the evolution of animal genomes, the evolution of plant genomes is indeed much more complicated. The complicated performance is introduced in lines 141-144, 626-628, and 654-659 of the manuscript. This sentence has been deleted (lines 81-82).

  1. d) line 92 - line 93: Are there any benchmarking studies for subcellular localization prediction tools?

We have not done research in this area. This article mainly analyzes the evolutionary history of APX gene in maize. Did not do very detailed research on gene function In order to make the prediction results more accurate, we made predictions using a variety of prediction software that can be found on the Internet.

  1. e) The sentence in line 96 should be removed, because it is very superficial.

This sentence has been deleted in line 99 of the manuscript.

3) line 136 - line 140 are introduction not methods. The sentence in line 141 needs to be rephrased. The authors still not explain HOW the search for candidate genes was performed. Which tool was used? Which parameters? Unfortunately, I cannot access the supplementary files. The files seems to be broken.

In lines 141-147 of the manuscript, we explained the complexity of plant genome evolution and the basis for selecting these species. Lee et al. [24] established a phylogenetic tree of 26 species, predominantly those with published genome sequences with known ancient whole genome duplications marked. The phylogenetic tree established by Lee et al. [24] is shown below. We selected species marked with pink, blue and green dots. In addition, in order to study the origin of the angiosperm APX, we also selected 8 other species (lines 153-158).

The sentence in line 141 has been rewritten (lines 146-147).

For the search of candidate genes in each species, we directly use the tools in the e! Ensembl Plants database to search. The method is, based on the previous references, to find the six APX genes of Arabidopsis thaliana in the database first. Then, use the tools in the database to find the orthologous genes of A. thaliana 6 genes in other species, and find the paralogous genes in other species. Finally, screen the found genes. The protein sequences encoded by the genes are too short, such as less than 150 amino acids. These genes are first excluded. The remaining protein sequences encoded by genes are downloaded and the APX domain prediction is performed on the Pfam website. Proteins with complete domains (corresponding genes) are retained. The details are explained in lines 166-175 and 249-258 of the manuscript. When submitting again, we will submit the "Protein Sequence File in supplementary materials" separately.

4) I cannot find the explanation for "Using an e-value of 10 as cutoff will return many unspecific hits. I would expect to see many more sequences in the phylogenetic tree. Were additional filter criteria applied? The authors might want to explain their choice of BLAST parameters. Also, which version of BLAST was used?". The authors should provide line numbers to make this easy to check.

The screening of the APX gene of the species "Chara braunii" is done on the NCBI website. In the NCBI website, select the blastp program, select the Chara braunii (taxid: 69332) database, and then enter the protein sequence of A. thaliana tAPX to complete the protein sequence alignment. The operation steps are as follows:

  1. After selecting the blastp program, enter the query sequence interface
  2. Select Chara braunii (taxid:69332) database
  3. Adjust the expected threshold
  4. The sequence alignment results are as follows. A total of 7 proteins were found in Chara braunii, and these protein sequences can be downloaded.
  5. The downloaded protein sequence size and domain are analyzed (as described above). It was finally confirmed that Chara braunii has 5 APX genes.

Considering that researchers in this area should be familiar with the use of these tools, the manuscript does not explain the operation steps in detail. The brief description is in lines 169-175 of the manuscript.

5) No, the nucleotide sequences has more signal. Aligning peptide sequences and translating into the nucleotide sequences can improve the resulting phylogenetic tree.

I quite agree with you. Due to the appearance of synonymous codes, each amino acid may correspond to multiple codons, so the nucleotide sequence has more signals. But it also makes the established evolutionary tree more complicated, and brings greater difficulties to analysis and judgment. Therefore, in order to facilitate analysis, protein sequence alignment is often used to build a phylogenetic tree. As mentioned earlier, our evolutionary study of zebrafish genes was to use protein sequence alignment to build a phylogenetic tree. In addition, the established phylogenetic tree cannot be used as the only evidence for judging the evolutionary history of genes. It also needs to combine the analysis results of gene splicing sites, replication events and positioning on chromosomes. The instructions are in lines 330-332, 396-403 of the manuscript.

6) Where was this addressed "It appears that the authors analysed peptide sequences, but are writing about genes in the next paragraph. Consistent terminology should be used clearly throughout the manuscript. Genes might give rise to different transcripts/peptides. How were these isoforms handled in this study?"?

I'm very sorry, because I don't master the language of English, there are some errors in the description of the manuscript. We have try to revised similar errors in the manuscript and are also eager to express our ideas accurately. What we analyze is the protein sequence encoded by gene. If a gene has multiple transcripts, the transcript with the longest protein sequence will be selected and the longest protein sequence will be downloaded. For instructions, please check 168-169 of the manuscript. Strictly speaking, it should be a polypeptide sequence (some proteins have multiple subunits). Considering that many references described it by using protein sequences, protein sequences are used in the manuscript.

In this manuscript, the subcellular localization of the different protein isoforms encoded by multiple transcripts of a gene has not been studied. However, in Figure 5 of the manuscript, the expression levels of different transcripts of 3 genes (APXa3, APXb1, APXc1.1) were tested (line 530).

7) ok

8) What is about the splice sites? Do all genes have the canonical GT-AG motif or are there any non-canonical splice sites conserved in this gene family? If the exon-intron structure is highly conserved, there might be some conservation inside the introns too.

The analysis of "splice site" in this manuscript is not an analysis of the conservative nucleotide sequence of "splice site". The analysis of the splicing site in this manuscript refers to the analysis of the phase of the introns at the splicing site and the number of nucleotides contained in each exon. Conserved exon/intron structures including exons with the same numbers of nucleotides as well as the conserved intron phases provide evidence for gene similarities and alternative characters independent of nucleotide or amino acid sequence(in lines 330-332). The structural information of genes can be obtained in e! Ensembl Plants. As shown below:

  1. Select transcripts, exons
  2. The "phase" of introns and the number of nucleotides in each exon

9) The authors are frequently using "replication", but they probably should use "duplication" (e.g. line 390). Why should replication time effect genome duplications?

"replication" has been replaced with "duplication" (line 416).

Considering that gene duplication may have both tandem duplication and local duplication. The copies of genes produced by the two duplication methods are on one chromosome, and the distance between the copies is close. Therefore, it is difficult to determine which of these two methods is used to complete gene duplication. So, the two methods are both marked (lines 16, 272, 281, etc.). For "Why should replication time effect genome duplications?", our wording is not accurate. We mean the effect of the frequency of genome duplication on the genome and the number of genes. The manuscript has been revised (lines 141-144).

10) How is the chromosomal distribution similar (line198)? The chromosome numbers differ between plant species. "Homologous" chromosomes in different species might not have matching numbers. The legend of Fig. 4 needs to explain what the different abbreviations mean and which chromosomes are displayed.

The explanation of "How is the chromosomal distribution similar (line198)?" is in lines 417-427 of the manuscript.

Various abbreviations have been explained in the legend in Figure 4 (line 435-443).

11) The authors performed only predictions of the subcellular localization. This needs to be clearly indicated as a prediction throughout the text (e.g. header of Table1).

Modification has been made on line 504.

12) What is the conclusion from the gene expression analysis? Is there a general picture? The font size in Fig. 5 appears very small and is not readable.

The conclusions of gene expression analysis are in lines 962-966 of the manuscript. There was a general picture of gene expression analysis in the "conclusion" of the original manuscript. We accept the reviewers’ suggestions and the “conclusions” in the revised manuscript have been simplified.

The biggest modification to the font of Figure 5 has been made. The figure in the manuscript can only be enlarged by stretching.

13) If candidate genes are missing from the microarray, the authors should use RNAseq data sets instead which are not restricted to certain gene sets. How does the clustering by gene expression compare to the phylogenetic relationship?

Due to the limited number of genes in the microarray used in the experiment we selected, no results were observed for few genes. If we can use the "RNAseq data set", it will be more helpful to study the functional differentiation of maize APX genes. As mentioned above, since this manuscript mainly establishes an angiosperm, especially maize, APX gene evolution model, the research on function is not the focus, and no more experimental data is selected for analysis.

The clustering of gene expression has no correspondence with the clustering of phylogeny.

14) The author speculate about gene loss. Have the authors checked that there are no additional (pseudo)genes in any of the Zea mays genomes? It is possible that some gene copies are lost in specific cultivars, but are still present in others.

As mentioned earlier, we have carried out APX gene identification on three rice varieties in e! Ensembl Plants and found that the types and numbers of APX genes in these three varieties are the same. Since there are no multiple maize varieties in the database, we have not identified APX genes of different maize varieties.

15) Orthologous genes are likely to have a similar function (line 372). However, it cannot be assumed that all orthologs will have the very same function. Especially when transferring functional annotations across a long phylogenetic distance, there is a substantial uncertainty involved. The authors might want to rephrase their sentence to better indicate this. Also, I strongly disagree with the statement that the inference of orthologs to A. thaliana would be impossible. The authors need to include additional species to cover the large phylogenetic gap. If the identification of orthologs is not feasible, it would be possible to identify at least orthogroups.

We have made changes, please check lines 728-731.

16)

  1. a) The authors should rephrase their discussion (originally line 376-line 394 and also the following paragraphs) to clearly indicate that these are assumptions derived from research on rice orthologs. It is important to clearly indicate that the function was not shown in Zea mays yet. This study only contains a phylogenetic analysis and expression analyses. However,t he actually function of the enzymes was not shown. How can the authors exclude that some of these genes are encoding non-functional products?

We have rewritten the discussion section. Based on the predicted results of the subcellular localization of the protein encoded by the maize APX gene and the specific tissue expression sites of the genes, combined with the research results of rice, the possible function of the maize gene was speculated. See 734-792, 828-865.

  1. b) line 565-566: "These results prove that the APX genes of class A are the oldest gene." There could be an older lineage which is absent from the investigated species. Therefore, conclusions should be phrased more carefully.

This conclusion is based on the genes of unicellular green algae in class A, which are the earliest APX genes. See line 740.

17/18)

  1. a) "Maize and rice orthologous genes have similar tissue expression patterns" This sentence in the conclusion appears to be conflicting with the authors' response "However, the expression of the eight APX genes of maize at different developmental stages of maize and the changes in gene expression levels under stress conditions have proved that the mechanisms of expression regulation of these genes are different."

I did not understand your opinion very well. My answer below may not be accurate. "Maize and rice orthologous genes have similar tissue expression patterns" means that each pair of maize and rice orthologous genes have similar tissue expression patterns. It does not mean that the eight genes in maize (or rice) have similar tissue expression patterns.

  1. b) The conclusion is inaccurate, because previous studies already addressed the APX evolution e.g. https://link.springer.com/article/10.1007/s00438-017-1413-2 and https://www.sciencedirect.com/science/article/pii/S0888754320300379. It appears that some relevant references might be missing here.

I'm very sorry, we did not find these 2 articles at the time of writing this manuscript. I downloaded these 2 articles according to the address you provided. Similar to the articles we have seen before, no evolutionary model of APX genes has been established, and genes cannot be named according to evolutionary relationships. The first article (2018) included Arabidopsis thaliana APX4 and its orthologous genes, thus clustering into 4 groups. The second article (2020) contains Arabidopsis thaliana APX4 and APX6 and their orthologous genes, thus clustering into 5 large groups. Studies have confirmed that the A. thaliana APX4 and APX6 and their orthologous genes have no APX activity [9], so they are not true APX genes. The current research on APX gene does have the following problems. For example, many studies are still classifying APX4 and APX6 and their orthologous genes as APX genes, so it is impossible to accurately determine the number of APX genes in each species. So far, no articles have been found to establish an angiosperm gene evolution model, clarify the evolutionary relationship of each member of the gene family, and make a reasonable name. Most of the articles classify the genes of class C into class I, mainly the A. thaliana genes named APX1 and APX2 in this group. But our evolutionary analysis found that these genes appeared last.

19) The main question that I see is "why are there so many gene copies?". The authors show different expression domains which could be the explanation. However, to conclude that this is the main reason, it would be necessary to check for other changes.

Sorry, I did not understand your comments well. Did you mean that there are so many APX genes in maize, and what is their specific functional differentiation determined through this study? Since we did not analyze the structure and function at the protein level, we did not verify the protein expression site through experiments. Therefore, the difference in the function of these isoforms cannot be confirmed at the protein level. However, in this article, by analyzing the results of published research and our predictions using a variety of software, combined with the results of the specific tissue expression of these genes, we have proposed a more accurate prediction of the subcellular localization of these isoforms. The proteins encoded by these genes are located in different subcellular locations, expressed in different tissues, and have different responses to stress. These results all indicate their functional differentiation. This provides information for further research.

20) line 304-line307 should be moved to the discussion. Although these questions cannot be completely answered, I think they could be addressed. It might not be possible to provide perfect answer, but it would be possible to predict e.g. the number of gene copies in an ancestor. I can see from the tree that there were at least 3 copies.

I think your suggestion is reasonable. But I think it is reasonable not to move this part of the content, which can lead to the following content more naturally and emphasize the importance of the following content (line 313- line 316).

We have also analyzed the evolution of other genes, and each type of gene may not have only one ancestral gene. Before the ancestors have undergone genome duplication, if a certain type of ancestral gene is copied through tandem duplication, then this type of gene may have two ancestral genes.

21) The method section generates the impression that these experiments would have been part of this study. If the experiments are described in the original studies, it should be sufficient to just cite them.

We have deleted this part of the content.

22) line 314-line319: How can the authors tell that the ancestral gene has not lost its introns in algae species?

At the bottom right of Figure 2 (line 349), there are 4 genes with only one exon. Specifically, they are the 6186 gene of OL (Ostreococcus lucimarinus), the 158C gene of CM (Cyanidioschyzon merolae), and the 24001 and 40001 genes of CC (Chondrus crispus).

23) What are "intron phases" (line 336)? Is this supposed to be the phase of coding sequences in the exons?

"Intron phases" refers to the relationship between the site of intron insertion and the codon in the gene coding region. The "phase" of introns has three manifestations, "0", "1" and "2". A codon is composed of 3 nucleotides. If the insertion site of an intron is between two codons, the phase of the intron is "0". If the insertion site of an intron is between the first and second nucleotides of a codon, the phase of the intron is "1". If the insertion site of the intron is between the second and third nucleotides of a codon, the phase of the intron is "2".

Minor comments:

line 329: Musa acuminate > Musa acuminate

It has been modified on line 356.

line 191 (and others): "cDNA sequences" > transcript sequences

It has been modified on line 199, 200.

line 314 and line 331: Physcomitrella paten > Physcomitrella patens

It has been modified on line 151, 268, 358.
